# LIFT: Learning to Fine-Tune via Bayesian Parameter Efficient Meta Fine-Tuning

**Minyoung Kim**[1] **& Timothy M. Hospedales**[1,2]
[1]Samsung AI Center Cambridge, UK          [2]University of Edinburgh, UK
mikim21@gmail.com                           t.hospedales@ed.ac.uk

## Abstract

We tackle the problem of parameter-efficient fine-tuning (PEFT) of a pre-trained large deep model on many different but related tasks. Instead of the simple but strong baseline strategy of task-wise independent fine-tuning, we aim to meta-learn the core shared information that can be used for unseen test tasks to improve the prediction performance further. That is, we propose a method for *learning-to-fine-tune* (LiFT). LiFT introduces a novel hierarchical Bayesian model that can be superior to both existing general meta learning algorithms like MAML and recent LoRA zoo mixing approaches such as LoRA-Retriever and model-based clustering. In our Bayesian model, the parameters of the task-specific LoRA modules are regarded as random variables where these task-wise LoRA modules are governed/regularized by higher-level latent random variables, which represents the prior of the LoRA modules that capture the shared information across all training tasks. To make the posterior inference feasible, we propose a novel SGLD-Gibbs sampling algorithm that is computationally efficient. To represent the posterior samples from the SGLD-Gibbs, we propose an online EM algorithm that maintains a Gaussian mixture representation for the posterior in an online manner in the course of iterative posterior sampling. We demonstrate the effectiveness of LiFT on NLP and vision multi-task meta learning benchmarks.

## 1 Introduction

Applied machine learning in most domains is increasingly dominated by a paradigm of fine-tuning models pre-trained on large-scale unlabelled data with self-supervised objectives (Bommasani et al., 2021). The consolidation of the community around this workflow has led to a flourishing of research in parameter-efficient fine-tuning (PEFT) methods (Ding et al., 2023; Hu et al., 2022). PEFT methods insert, or select, a small set of learnable parameters with the pre-trained model and update only these on the task of interest, while keeping the rest of the model frozen. The aim is to benefit from the representation provided from a large pre-trained model, while only learning a small number of parameters to avoid overfitting and keep the marginal parameter storage cost per task low. Popular PEFT methods such as LoRA (Hu et al., 2022), nowadays often underpin flagship new capabilities in NLP (Ding et al., 2023), vision and multi-modal (Liu et al., 2024) AI.

The PEFT paradigm's efficacy is due to effective knowledge transfer from the upstream pre-trained model to the downstream task of interest. However, in most workflows each downstream task is solved independently, and there is no exploitation of knowledge transfer between the growing array of supervised tasks that are tackled downstream. This has led some recent studies to ask whether PEFT modules can be re-used across multiple downstream tasks in order to further improve performance? For example LoRA-HUB (Huang et al., 2023) proposes to learn new tasks as linear combinations of old tasks' LoRA modules, and LoRA-Retriever (Zhao et al., 2024b) proposes to solve new tasks as mixtures of old tasks' LoRAs.

In this paper we propose to tackle the cross-task PEFT knowledge transfer problem from the perspective of Bayesian meta-learning. Meta-learning (Hospedales et al., 2021) aims for "learning how to learn" by jointly learning multiple-source tasks, and extracting some knowledge of "how to learn" that enables improved learning of the target tasks. *Bayesian* meta-learners extract such task-agnostic knowledge in the form of Bayesian priors. Rather than independently learning each source tasks' PEFT module and recombining them for a new target task (Huang et al., 2023; Zhao et al., 2024b), we

*jointly* learn the source tasks with a hierarchical Bayesian model, and extract a prior over PEFTs to improve target task learning. More specifically, we learn a mixture prior over PEFT modules, which can be considered as extracting a small set of prototypical latent task templates from the source tasks. This results in a more compact set of transferred knowledge compared to the exhaustive library-based methods. Meanwhile, each target task is effectively regularised by the most relevant latent tasks, which effectively boosts target task performance while combating overfitting.

Our approach, termed "Learning to Fine-Tune" (LiFT) can be efficiently implemented using the SGLD-Gibbs sampler for both source and target task learning, making it both theoretically elegant and practically effective. The novel SGLD-Gibbs sampling algorithm is one of our main technical contributions, which is computationally efficient, making the posterior inference feasible with many training tasks. Another technical contribution is that we propose an online EM algorithm that efficiently maintains a Gaussian mixture representation of the posterior samples from the SGLD-Gibbs posterior in an online manner in the course of iterative posterior sampling.

On a range of NLP and Vision tasks, our experimental results show that LiFT outperforms both classic meta-learning methods (MAML (Finn et al., 2017a), Reptile (Nichol et al., 2018), etc.) and previous Bayesian meta-learning algorithms (e.g., BMAML (Yoon et al., 2018) and ABML (Ravi & Beatson, 2019a)) by large margin. Compared to the latter, we have also observed improved uncertainty quantification thanks to the hierarchical Bayesian nature of the proposed method. Our meta learning approach also exhibits comparable or often superior performance to recent library-based approaches, including LoRA-Retriever (Zhao et al., 2024b) and the model-based PEFT clustering method (Ostapenko et al., 2024) on cross-task benchmarks.

## 2 METHODOLOGY

### 2.1 PROBLEM SETUP

We have many training tasks $\mathcal{T}_1, \mathcal{T}_2, \ldots, \mathcal{T}_N$ where each task $\mathcal{T}_i$ only reveals training data $D_i \sim \mathcal{T}_i$. We assume availability of a strong pre-trained deep (foundation) model parameters $\theta^0$ which is powerful and useful for novel downstream tasks. We follow the parameter-efficient fine-tuning (PEFT) strategy: freezing $\theta^0$ and introducing/learning the extra adapter modules (e.g., LoRA (Hu et al., 2022)). The parameters of these extra modules for the task $\mathcal{T}_i$ are denoted by $\theta_i^a$. That is, $(\theta_i^a, \theta^0)$ forms a model for the task $\mathcal{T}_i$ where only $\theta_i^a$ is to be learned with $D_i$, and $\theta^0$ frozen. Although our approach is not restricted to the type of the PEFT adapter architecture employed, our experiments focus on the LoRA PEFT for its popularity and strong downstream performance. At the downstream test time, we are given training examples $D_*$ sampled from an unseen task $\mathcal{T}_*$ so that we can do test-time adaptation/fine-tuning of the meta-trained model with $D_*$.

### 2.2 A HIERARCHICAL BAYESIAN MODEL FOR META-PEFT

In our Bayesian model design, we assume that each task (data) $\mathcal{T}_i (D_i)$ is generated/governed by the random variables $\theta_i \overset{\text{def}}{=} (\theta_i^a, \theta^0)$. Recall that $\theta_i^a$ is the LoRA parameters (random variables) specific to the task $i$, and $\theta^0$ is the base model parameters (constant). We introduce a higher-level random variable $\phi$ that regularizes *all* task-wise variables $\{\theta_i^a\}_{i=1}^N$ ($N$ is the number of training tasks). We can think of $\phi$ as some core LoRA parameter values that may be useful for all different types of tasks (shared across tasks). Also, since $\phi$ is a random variable, it can take multiple different values, presumably one of the several key prototypes or representative LoRA values *a posteriori* (i.e., if conditioned on the meta training set). See Fig. 1 for the graphical models.

Assuming i.i.d. generation of $\theta_i^a$ given $\phi$, the prior distribution becomes:

$$p(\phi, \{\theta_i^a\}_{i=1}^N) = p(\phi) \cdot \prod_{i=1}^N p(\theta_i^a | \phi) \tag{1}$$

We specifically consider the Gaussian priors: $p(\theta_i^a | \phi)$ is assumed to be centered at $\phi$, and $p(\phi)$ to be 0-centered, considering that 0 LoRA modules essentially amounts to the pre-trained model itself with no adapter module added. That is,

$$p(\phi) = \mathcal{N}(\phi; 0, \sigma^2 I), \quad p(\theta_i^a | \phi) = \mathcal{N}(\theta_i^a; \phi, \beta^2 I) \tag{2}$$

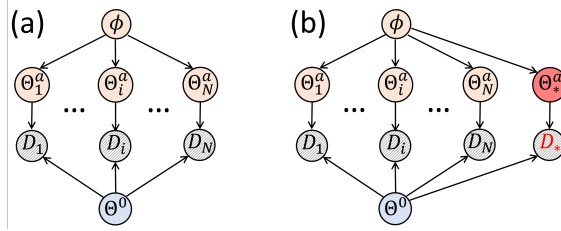

Figure 1: Our graphical models for (a) meta training and (b) meta test. Gray-filled nodes (data) indicate evidences, orange/red-filled nodes are latent random variables, and the blue node (base model parameters) is constant. In meta test (b), we have the latent LoRA variable $\theta_*^a$ associated with the observed test data $D_*$ (indicated by red).

where $\sigma^2$ and $\beta^2$ are fixed (user-chosen) variances.

For the likelihood, each LoRA parameters $\theta_i^a$, together with the frozen pre-trained parameters $\theta^0$, is associated with the data $D_i$ through the underlying loss function $l_i(\theta_i^a, \theta^0; D_i)$. More specifically,

$$p(D_i|\theta_i^a) \propto \exp(-l_i(\theta_i^a, \theta^0; D_i)) \tag{3}$$

Note that the loss function $l_i$ can be task specific (e.g., in the NLP cases, $l_i$ can be the cross-entropy loss for classification tasks, while it can be the next token prediction cross-entropy loss for the text generation tasks such as text summarization).

## 2.3 POSTERIOR INFERENCE BY SGLD-GIBBS SAMPLING

From our model definition, the posterior distribution becomes:

$$p(\phi, \{\theta_i^a\}_{i=1}^N | \{D_i\}_{i=1}^N) \propto p(\phi) \cdot \prod_{i=1}^N p(\theta_i^a|\phi)p(D_i|\theta_i^a) \tag{4}$$

Among several options for the approximate posterior inference (e.g., variational inference, MCMC), we adopt the stochastic-gradient MCMC sampling (Welling & Teh, 2011), and combine it with the Gibbs sampling (Gelfand & Smith, 1990; MacKay, 2003) for the computational efficiency, which will be clear in our explanation below.

First, applying the stochastic-gradient Langevin dynamics (SGLD) sampler, leads to the following state transition where the state here being comprised of all random variables $\phi$ and $\{\theta_i^a\}_{i=1}^N$:

$$[\phi, \theta_1^a \dots \theta_N^a] \leftarrow [\phi, \theta_1^a \dots \theta_N^a] + \frac{\eta}{2}\nabla_{\phi,\{\theta_i^a\}_{i=1}^N} \log p(\phi, \{\theta_i^a\}_{i=1}^N | \{D_i\}_{i=1}^N) + \sqrt{\eta} \cdot z \tag{5}$$

where $\eta$ is the step size, $z \sim \mathcal{N}(0, I)$, and the derivative is computed with respect to $\phi$ and all $\{\theta_i^a\}_{i=1}^N$. A main benefit of adopting SGLD is that we can avoid the difficult normalizing constant in (4) since the derivative of the normalizing constant with respect to $(\phi, \{\theta_i^a\}_{i=1}^N)$ in (5) becomes 0. This simplifies the SGLD recurrence as:

$$[\phi, \theta_1^a \dots \theta_N^a] \leftarrow [\phi, \theta_1^a \dots \theta_N^a] + \frac{\eta}{2}\nabla \left( \log p(\phi) + \sum_{i=1}^N \left( \log p(\theta_i^a|\phi) + \log p(D_i|\theta_i^a) \right) \right) + \sqrt{\eta} \cdot z \tag{6}$$

This can be written equivalently as:

$$\phi \leftarrow \phi + \frac{\eta}{2}\nabla_\phi \left( \log p(\phi) + \sum_{i=1}^N \log p(\theta_i^a|\phi) \right) + \sqrt{\eta} \cdot z_\phi \tag{7}$$

$$\theta_i^a \leftarrow \theta_i^a + \frac{\eta}{2}\nabla_{\theta_i} \left( \log p(\theta_i^a|\phi) + \log p(D_i|\theta_i^a) \right) + \sqrt{\eta} \cdot z_{\theta_i^a} \quad (i = 1, \dots, N) \tag{8}$$

According to the SGLD theorem (Welling & Teh, 2011; Raginsky et al., 2017; Xu et al., 2018; Zou et al., 2021), under mild conditions, the recurrence (6) is guaranteed to converge to the samples from the posterior $p(\phi, \{\theta_i^a\}_{i=1}^N | \{D_i\}_{i=1}^N)$.

However, running each recurrence step is computationally very demanding since one has to retain the states and gradients for all $\{\theta_i^a\}_{i=1}^N$ in the (GPU) memory even if each adapter module $\theta_i^a$ takes up a

relatively small memory footprint. To avoid this issue, we propose a novel *SGLD-Gibbs sampling* which combines the Gibbs sampler (Gelfand & Smith, 1990; MacKay, 2003) and SGLD sampling.

• **SGLD-Gibbs Sampling.** Following the Gibbs sampling practice in MCMC, we can reform the chain by randomly picking a subset of the state at each step, and do MCMC progression. As long as all variables in the state are visited sufficiently many times and frequently, it is known that the new chain also converges to the target posterior distribution (Gelfand & Smith, 1990). In our case, we pick a single $\theta_i^a$ in each SGLD step (6) sequentially in the round-robin manner. Since $\phi$ is the ultimate shared information we aim to learn/infer during meta training, we always choose $\phi$ in the Gibbs sampling step, which turns out to lead to faster convergence in practice. Another computational bottleneck is the sum of the gradients $\nabla_\phi \sum_i \log p(\theta_i^a | \phi)$ over all $i = 1, \ldots, N$ for each $\phi$ update in (7). To remedy this issue, we introduce an auxiliary variable $J$ that maintains this sum of the gradients, and we let it asynchronously updated (Eq. (11)): at each iteration for task $i$, subtracting the old gradient for $i$ from $J$ to approximate the cavity sum $\sum_{i' \neq i} \nabla_\phi \log p(\theta_{i'}^a | \phi)$ and adding a new gradient $\nabla_\phi \log p(\theta_i^a | \phi)$ to $J$. More specifically, each SGLD-Gibbs step consists of:

$$\phi \leftarrow \phi + \frac{\eta}{2} \nabla_\phi \log p(\phi) + \frac{\eta}{2} J + \sqrt{\eta} \cdot z_\phi \tag{9}$$

$$\theta_i^a \leftarrow \theta_i^a + \frac{\eta}{2} \nabla_{\theta_i^a} \left( \log p(\theta_i^a | \phi) + \log p(D_i | \theta_i^a) \right) + \sqrt{\eta} \cdot z_{\theta_i^a} \tag{10}$$

$$J \leftarrow J - \nabla_\phi \log p(\theta_i^{a \text{(old)}} | \phi) + \nabla_\phi \log p(\theta_i^{a \text{(new)}} | \phi) \tag{11}$$

for $i = 1, 2, \ldots, N, 1, 2, \ldots, N, \ldots$. The variable $J$ maintains $\nabla_\phi \sum_{i=1}^N \log p(\theta_i^a | \phi)$ in (7) approximately, which remains asynchronously up-to-date due to (11). We initialize $J$ to 0. The justification[1] of the SGLD-Gibbs sampling is empirically verified on some toy experiment in Appendix A where we show the convergence of the proposed SGLD-Gibbs sampling to the true posterior.

As will be seen in Sec. 2.4 and Fig. 1(b), we only need the posterior of the higher-level $p(\phi | \{D_i\}_{i=1}^N)$ during the downstream meta test stage. Hence, we only maintain the posterior samples $\{\phi^{(m)}\}_{m=1}^M$ from the SGLD-Gibbs recurrence (9–11) after some burn-in period. The next issue is how we represent the posterior samples succinctly. One popular choice is the Gaussian approximation (i.e., summarizing $\{\phi^{(m)}\}_{m=1}^M$ by their first- and second-order moments). However, we aim to enrich it by a *mixture of Gaussians* to better approximate the true posterior that is inherently a multi-modal distribution. In the implementation, we save the latest parameter samples $\theta_i^a$s either in the GPU memory if space is available or in the disk otherwise. In the latter case we load from the disk the latest $\theta_i^a$ for the current Gibbs subset $i$ in the SGLD-Gibbs iteration.

• **Online-EM Mixture for Posterior Approximation.** A main challenge in representing the $M$ posterior samples $\{\phi^{(m)}\}_{m=1}^M$ as a mixture of Gaussians (e.g., via the famous Expectation-Maximization (EM) algorithm (Dempster et al., 1977)), is that we may not store all those $M$ samples in one place in batch due to the memory overhead[2]. As a remedy, we propose a novel *online-EM mixture maintenance algorithm*[3]: We only access $\phi^{(m)}$ one at a time without revisit, and update the mixture in an online fashion. We aim to approximate the posterior by an order-$K$ Gaussian mixture:

$$p(\phi | \{D_i\}_{i=1}^N) \approx \sum_{j=1}^K \alpha_j \mathcal{N}(\phi; \mu_j, \Sigma_j). \tag{12}$$

Upon observing a posterior sample $\phi^{(m)}$, our online EM algorithm updates the parameters of the mixture by the following equations (Detailed derivations can be found in Appendix B):

- (E-step) Compute the following component assignment probabilities at the current mixture:

$$q_j = \frac{\alpha_j \mathcal{N}(\phi^{(m)}; \mu_j, \Sigma_j)}{\sum_{j'=1}^K \alpha_{j'} \mathcal{N}(\phi^{(m)}; \mu_{j'}, \Sigma_{j'})} \quad (j = 1, \ldots, K) \tag{13}$$

---

[1] Our SGLD-Gibbs, without the asynchronous update with $J$, has a transition kernel in MCMC defined to be a composition of the SGLD kernel and the Gibbs kernel that are both known to converge to the target distribution. Hence SGLD-Gibbs should converge to the target distribution in theory.

[2] For instance, the number of posterior samples ($M$) can be as large as the number of whole training steps.

[3] There exist several online/sequential EM algorithms that exploit the recursive structure of the EM in manners similar to ours (Appendix D for details). These prior methods may be employed in place of our proposed one.

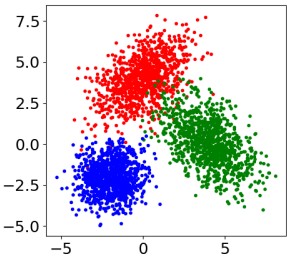 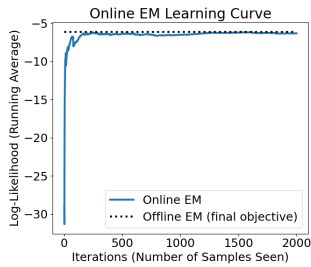

Figure 2: (Toy experiment for online EM) (Left) Data generated from a true mixture distribution. (Right) Online EM learning curve.

- (M-step) Update the mixture parameters as follows (for $j = 1, \ldots, K$):

$$\alpha_j \leftarrow \frac{n_j + q_j}{1 + \sum_{j'} n_{j'}}, \ \mu_j \leftarrow \frac{n_j \mu_j + q_j \phi^{(m)}}{n_j + q_j}, \ S_j \leftarrow \frac{n_j S_j + q_j \phi^{(m)} \phi^{(m)\top}}{n_j + q_j}, \ \Sigma_j = S_j - \mu_j \mu_j^\top$$

(followed by the update: $n_j \leftarrow n_j + q_j$) \hfill (14)

One can interpret the formulation above as a one EM step applied to the current MoG estimate. Due to the high dimensionality of $\phi$ we often restrict $\Sigma$ (and $S$) to be diagonal or spherical in practice. Also, the initial $\{n_j\}_{j=1}^K$ can be seen as the prior strength in the mixture estimation, and we find that initial $n_j = 100$ often works well in practice.

To justify our online mixture estimation method, we have conducted a proof-of-concept toy experiment. For the data we devise a true mixture distribution with three Gaussian components in 2D. From this true model, we generate 3000 samples (1000 samples per component) as shown in Fig. 2(Left). The offline EM which accesses all 3000 samples at once converges to its (local) minimum (data log-likelihood objective) $-6.15$ after 16 EM iterations (i.e., 16 epochs). Our online EM algorithm converges even faster to its minimum objective $-6.07$ (the running averaged objective $-6.17$) after seeing only about 2000 samples, which corresponds to $2/3$ epoch. The results are shown in Fig. 2(Right). We also compute the KL divergences between the true mixture and the EM-learned mixtures: $\text{KL}(p^{\text{true}} || p^{\text{offline}}) = 0.0781$ while $\text{KL}(p^{\text{true}} || p^{\text{online}}) = 0.0686$. This signifies the effectiveness of the proposed online EM algorithm.

## 2.4 META TEST AS BAYESIAN INFERENCE

At test time, for the given training data $D_*$ from the unseen test task $\mathcal{T}_*$, we would ultimately like to infer the posterior distribution of the LoRA module for the target task, denoted by $\theta_*^a$. That is, $p(\theta_*^a | D_*, \{D_i\}_{i=1}^N)$ where the evidence now encompasses all the training data $\{D_i\}_{i=1}^N$ and the test data $D_*$. This posterior can be derived as follows:

$$p(\theta_*^a | D_*, \{D_i\}_{i=1}^N) \ \propto \ p(\theta_*^a, D_* | \{D_i\}_{i=1}^N) = \int p(\theta_*^a, D_*, \phi | \{D_i\}_{i=1}^N) \, d\phi \tag{15}$$

$$= \int p(D_* | \theta_*^a) \, p(\theta_*^a | \phi) \, p(\phi | \{D_i\}_{i=1}^N) \, d\phi \ = \ p(D_* | \theta_*^a) \cdot \mathbb{E}_{p(\phi | \{D_i\}_{i=1}^N)}[p(\theta_*^a | \phi)] \tag{16}$$

From (15) to (16), we use the conditional independence from our model (Fig. 1(b)), namely $\theta_*^a \perp \{D_i\}_{i=1}^N | \phi$, that is, $\theta_*^a$ and $\{D_i\}_{i=1}^N$ are conditionally independent given $\phi$. Using our Gaussian mixture approximation of $p(\phi | \{D_i\}_{i=1}^N)$ in (12), the expectation in (16) can be written as:

$$\mathbb{E}_{p(\phi | \{D_i\}_{i=1}^N)}[p(\theta_*^a | \phi)] \approx \sum_{j=1}^K \alpha_j \mathbb{E}_{\mathcal{N}(\phi; \mu_j, \Sigma_j)}[p(\theta_*^a | \phi)] = \sum_{j=1}^K \alpha_j \mathcal{N}(\theta_*^a; \mu_j, \Sigma_j + \beta^2 I) \tag{17}$$

where in the latter equality of (17), we use the Gaussian identity with our Gaussian prior $p(\theta_*^a | \phi) = \mathcal{N}(\theta_*^a; \phi, \beta^2 I)$ from (2).

Since we have derived the posterior $p(\theta_*^a | D_*, \{D_i\}_{i=1}^N)$ up to constant, we can take samples from it by running the SGLD sampler. (Note that this is test-time SGLD sampling apart from the one for training.) More specifically, we repeat the following recurrence:

$$\theta_*^a \leftarrow \theta_*^a + \frac{\eta}{2} \nabla_{\theta_*^a} \left( \log p(D_* | \theta_*^a) + \log \sum_{j=1}^K \alpha_j \mathcal{N}(\theta_*^a; \mu_j, \Sigma_j + \beta^2 I) \right) + \sqrt{\eta} \cdot z \tag{18}$$

Intuitively, each step of (18) can be seen as an SGD step (with the added SGLD noise) for the penalized objective function: ① *loss on $D_*$* + ② *penalty of deviating from one of the (scale-adjusted) prototypes $\{\mu_j\}_{j=1}^K$ scaled by $\{\Sigma_j\}_{j=1}^K$*, where these prototypes were learned from meta training.

## 3 RELATED WORK

**General multi-task meta learning.** The gradient-based adaptation technique MAML (Finn et al., 2017b) is recognized as one of the most popular episodic meta learning algorithms due to its empirical success and known theoretical support (Chen & Chen, 2022). Several variants have been proposed to enhance MAML's performance as well as addressing its computational overhead, including the L2 regularization technique (Rajeswaran et al., 2019), the first-order approximation (Nichol et al., 2018), and probabilistic extension (Finn et al., 2018). Bayesian extensions to MAML (Finn et al., 2018; Yoon et al., 2018; Ravi & Beatson, 2019b; Nguyen et al., 2020) are highly related to our work, however, they are weak in several aspects. For instance, BMAML (Yoon et al., 2018) is not a hierarchical Bayesian model, but turns MAML's gradient-based *deterministic* adaptation steps to a *stochastic* counterpart using the particle-based stochastic ensemble-based adaptation steps. ABML (Ravi & Beatson, 2019a) relies on the variational inference, and resorts to a Gaussian posterior approximation.

**PEFT/LoRA adapter mixture/ensemble techniques.** Especially in the NLP domain with emphasis on cross-task adaptation (Ye et al., 2021), a large number of techniques have been proposed. Considering the large scale of the backbone networks (e.g., LLMs), many recent approaches aim to focus on adapting relatively smaller PEFT modules instead of the full backbone adaptation as in Ye et al. (2021). As it is overwhelming to carry out all of the literature reviews in this paper, we just highlight here a few works that are the most relevant to our method, including Meta-ICL (Min et al., 2022), mixtures-of-LoRAs (Feng et al., 2024; Asadi et al., 2024), LoRA Retriever (Zhao et al., 2024a), and model-based clustering method (Ostapenko et al., 2024). We leave the discussions on these works and relation to our approach in Appendix D.

**Modulation-based meta learning** & **Online EM algorithms.** There have been several works that consider meta-learning by updating relatively few parameters (modulation). There have been several prior works on the online EM methods in the literature. Although we find that none of them are not identical to the one proposed in this paper, most are similar in nature to ours in that the recursive structure of the EM is exploited. The discussions on these works can be found in Appendix D.

## 4 EXPERIMENTS

We test our LiFT algorithm on three benchmark datasets from NLP and vision for the cross-task PEFT transfer learning problem: i) **Cross-Fit** text-to-text generation NLP problem; ii) **VTAB** image classification/prediction vision problem; and iii) **Shakespeare** next word prediction NLP problem. The details of the experimental settings are discussed in the subsequent sections, Sec. 4.1, Sec. 4.2 and Sec. 4.3. We also conduct uncertainty quantification in Sec. 4.4 and ablation study in Sec. 4.5.

### 4.1 CROSS-FIT NLP BENCHMARK

Following the seminal work from Ye et al. (2021) on the cross-task NLP benchmark, we mostly follow their experimental design and task grouping ontology. There are 160 NLP tasks[4] in the text-to-text format. Each task consists of train/dev/test sets, where train and dev datasets contain 32 examples, and test datasets have the rest. Inspired by the task ontology introduced in (Ye et al., 2021), we consider 7 task splits for cross-task meta learning: `CLS-45`, `CLS-23`, `CLS-0`, `Multiple-Choice-Q/A`, `MRC-Q/A`, `NLI` and `Paraphrasing`. The first three contains all classification tasks for meta test where the meta train set consists of either all classification tasks (`CLS-45`), all non-classification tasks (`CLS-0`), or half-and-half (`CLS-23`). For the last four splits, we use the held-out tasks as a meta train set, e.g., in `MCQA`, all non-MCQA tasks are used in the meta training stage, while all MCQA tasks form the meta task set. Please see Fig. 5–6 in Appendix for details of the task splits.

---

[4]Since some datasets are outdated or license protected, not all tasks were available at the time of our experimental study. There are 23 tasks failed to download, and we only manage to use 137 tasks in this study.

Due to the diversity of the tasks, the related metrics are different (e.g., accuracy, F1 score, ROUGE-L, or EM scores). Even though a heuristic solution of reporting the average relative gain (ARG) was introduced in (Ye et al., 2021), we find that this metric is sensitive to the score ranges, and biased to those tasks for which the baseline model has near-0 scores. Hence we stick to the simple average of the metrics over all test tasks regardless of the metric types.

The results are summarized in Table 1, especially our proposed LiFT at the last rows against the first group of competing methods titled "Baselines / Meta learners". We use the BART-base encoder-decoder model (Lewis et al., 2020) as the main backbone with rank-4 LoRA for all competing methods. In our LiFT we vary the posterior mixture order $K = 1, 3, 5$. See Fig. 3 for the impact of the mixture order compared to the competing counterparts. The two baselines are: "No Meta Train" that simply ignores the meta train set, and only fine-tune the LoRAs on the training set of each meta test task; "Union Train" is a standard multi-task baseline where it combines all meta training tasks into a big single union dataset, and trains the LoRAs on the union set. At meta test time it performs PEFT starting from the trained model as a initial model. The four meta learners (MAML (Finn et al., 2017b), First-Order-MAML, i-MAML (Rajeswaran et al., 2019) and Reptile (Nichol et al., 2018)) are the most popular meta learning algorithms, for instance, MAML learns the best initial model with a one-step inner loop iteration in its reverse-mode hyper-gradient computation with the meta train tasks. It is evident that our LiFT significantly outperforms all these baselines and meta learners.

• **Comparison with Bayesian Meta Learning Methods.** In the second group of Table 1 titled "Bayesian", we compare previous Bayesian meta learning algorithms including Bayesian MAML (BMAML) (Yoon et al., 2018) and Amortized Bayesian Meta Learning (ABML) (Ravi & Beatson, 2019a) with our LiFT. Unlike our method that is based on the mixture posterior estimate, ABML's Gaussian posterior and the ensemble-based (with $M$ particles) BMAML fall short.

• **Comparison with Mixture of MAMLs.** A mixture version of MAMLs amounts to maintaining a set of $K$ initial parameters to be optimized, instead of a single initial parameters in the original MAML. For the iMAML mixture, we maintain the $K$ L2 regularizers in the inner optimization. The main difference of our LiFT from iMAML/MAML mixtures is that we use the Bayesian inference for training (SGLD-Gibbs + online EM) and testing (SGLD), while the latter approaches use numerically less stable implicit function theorem (Lorraine et al., 2020) or reverse-mode differentiation. Experimental results in the third group of Table 1 (titled "Mixtures") support our claim.

• **Comparison to LoRA Mixture Strategies.** Several recent multi-task LoRA mixture methods adopt the strategy of: i) task-wise LoRA training to form a LoRA zoo; and ii) at test time either use routing mechanism (MoE) or mixing the LoRA parameters, either all LoRAs in the zoo or selectively most relevant ones. LoRA Retriever (Zhao et al., 2024a) selects top-$K$ relevant LoRAs using the cosine similarity, and combine those $K$ LoRAs by either *mixture* or *fusion* operations. Another recent strategy is the model-based clustering (MBC-$\mu$) (Ostapenko et al., 2024), where they perform the k-means clustering on the task-wise trained LoRAs (after some PCA dimensionality reduction). In the fourth group of Table 1 titled "LoRA Mixtures", we contrast with the performance of these methods. They usually underperform our LiFT on this cross-task dataset/experiment.

## 4.2 VTAB VISION BENCHMARK

Next we test our LiFT on the cross-task vision problem formed using the VTAB-1K benchmark (Zhai et al., 2019), which is comprised of 19 different image datasets for image classification and prediction. These datasets exhibit highly diverse domains/aspects/conditions including: different image acquisition (e.g., from standard imaging to special purpose remote sensing or medical imaging), different objects/concepts (e.g., generic, fine-grained, or abstract concepts), and different prediction types (object recognition, counting, or pose estimation). Each of the 19 datasets consists of $1K$ training examples, and we use the conventional splits of train $80\%$ and validation $20\%$.

To form a cross-task problem for testing our meta fine-tuning model, we partition the 19 datasets into three groups along their domains. They are: `natural` (7 datasets), `special` (4), and `structured` (8). For instance, "Cifar100" and "Caltech101" belong to the `natural` group since their images are from natural scenes, while "Clevr-count" and "Dsprites-loc" are grouped in the `structured` group as they mainly contain shaped/structured objects. We then construct train/test task splits for each group (e.g., `natural`) by having datasets that do not belong to the group set as train tasks and those in the group as test tasks. The splits of tasks are shown at the head of Table 2.

Table 1: Cross-Fit cross-task transfer learning results with different task splits. For each task split, we report the metric scores averaged over test tasks (↑). See texts for details.

| | | | Classification | | | Non-classification | | | | Avg |
|---|---|---|---|---|---|---|---|---|---|---|
| | | | CLS-45 | CLS-23 | CLS-0 | MCQA | MRC | NLI | Para. | |
| Baselines / Meta learners | No Meta Train | | 57.34 | 57.34 | 57.34 | 28.61 | 21.87 | 53.08 | 51.49 | 46.72 |
| | Union Train | | 57.25 | 53.59 | 50.87 | 28.55 | 26.21 | 52.84 | 53.13 | 46.06 |
| | MAML | | 52.68 | 58.60 | 52.73 | 29.38 | 25.13 | 54.08 | 55.30 | 46.84 |
| | FO-MAML | | 59.38 | 55.25 | 51.84 | 25.70 | 27.21 | 54.90 | 51.20 | 46.50 |
| | i-MAML | | 59.61 | 57.91 | 58.51 | 28.53 | 26.12 | 49.79 | 55.65 | 48.02 |
| | Reptile | | 57.59 | 57.84 | 54.62 | 29.03 | 25.92 | 55.07 | 55.42 | 47.93 |
| Bayesian | BMAML | $M=5$ | 56.89 | 57.78 | 53.30 | **30.47** | 27.87 | 54.45 | 54.45 | 47.89 |
| | ABML | | 56.02 | 55.71 | 50.95 | 30.33 | 28.44 | 53.49 | 55.99 | 47.28 |
| Mixtures | MAML-Mix | $K=5$ | 54.02 | 60.13 | 54.16 | 29.97 | 26.57 | 55.08 | 55.97 | 47.99 |
| | i-MAML-Mix | $K=5$ | 60.13 | 59.36 | 59.24 | 29.03 | 26.47 | 53.48 | 56.90 | 49.23 |
| LoRA Mixtures | Retriever | $K=5$ | 55.08 | 53.04 | 54.09 | 28.31 | 21.31 | 51.67 | 52.51 | 45.14 |
| | Mixture | $K=$all | 51.68 | 54.57 | 55.32 | 27.92 | 21.33 | 51.68 | 53.03 | 45.08 |
| | Retriever | $K=5$ | 57.91 | 54.93 | 55.51 | 28.31 | 21.43 | 51.54 | 54.72 | 46.34 |
| | Fusion | $K=$all | 55.30 | 56.01 | 54.74 | 28.36 | 21.42 | 51.08 | 53.85 | 45.82 |
| | MBC-$\mu$ | $K=5$ | 55.03 | 55.25 | 55.44 | 28.30 | 21.36 | 52.10 | 53.79 | 45.90 |
| | | $K=10$ | 55.39 | 55.95 | 57.13 | 28.40 | 21.39 | 51.36 | 52.05 | 45.95 |
| Ours | LiFT | $K=5$ | **64.12** | **63.87** | **61.90** | 30.37 | **28.69** | **56.93** | **57.39** | **51.90** |

Figure 3: Impact of the model orders on the Cross-Fit benchmark. We vary the model order $K$ or $M$ to $1, 3, 5$ in the X-axis. It is the mixture order in our LiFT, MAML-Mix, and i-MAML-Mix; the number of particles in BMAML; the number of relevant LoRAs in Retriever-Mixture and Retriever-Fusion.

The results are summarized in Table 2. For all competing methods, we use the ViT-B/16 model (Dosovitskiy et al., 2021) pre-trained with ImageNet (Deng et al., 2009). For the PEFT module, the rank-4 LoRA adapters are added to the ViT where we put adapters to the Q/K/V embedding matrices as well as the output matrices. For our LiFT, we vary the posterior mixture order $K=1, 2, 3$. For the competing meta learning methods (MAML and its variants), we use a single inner step for computational tractability. The hyperparameters and experimental details can be found in Appendix C. The result indicates that our method, across all task groups, consistently outperforms the competing meta learning methods, the multi-task baselines, and the ProtoNet (Snell et al., 2017) and its memory-efficient full-batch version ProtoNet-LITE (Bronskill et al., 2021). This signifies the efficacy of the proposed hierarchical Bayesian meta PEFT algorithm on the vision cross-task transfer learning setting.

## 4.3 SHAKESPEARE DATA NEXT WORD PREDICTION

From the LEAF benchmark (Caldas et al., 2019), we collect Shakespeare play lines where we treat each character as a personalized task. Then the lines played by each character are regarded as data for

Table 2: VTAB-1K results.

| | Non-natural → Natural | | | | | | | | Non-special → Special | | | | | Non-structured → Structured | | | | | | | | |
|---|---|---|---|---|---|---|---|---|---|---|---|---|---|---|---|---|---|---|---|---|---|---|
| Method | Cifar100 | Caltech101 | DTD | Flower102 | Pets | SVHN | Sun397 | AVERAGE | Camelyon | EuroSAT | Resisc45 | Retinopathy | AVERAGE | Clevr-Count | Clevr-Dist | DMLab | KITTI | dSpr-Loc | dSpr-Ori | sNORB-Azim | sNORB-Ele | AVERAGE |
| No Meta Train | 63.0 | 91.1 | 67.8 | 98.5 | 90.3 | 83.9 | 51.4 | 78.0 | 84.5 | 94.6 | 83.8 | 74.4 | 84.3 | 70.0 | 66.2 | 45.0 | 74.8 | 71.5 | 46.7 | 25.7 | 35.2 | 54.4 |
| Union Train | 65.3 | 92.0 | 68.7 | **99.4** | 90.2 | 83.9 | 53.8 | 79.0 | 84.0 | 95.0 | 84.3 | 74.9 | 84.5 | 71.4 | 67.5 | 45.8 | 75.4 | 72.2 | 46.7 | 25.6 | 36.7 | 55.2 |
| MAML | 64.1 | 92.0 | 69.7 | **99.4** | 90.7 | 84.3 | **56.9** | 79.6 | 84.9 | 94.6 | 86.0 | 74.9 | 85.1 | 69.7 | 68.4 | 46.7 | 75.5 | 74.5 | 47.1 | 25.6 | 36.8 | 55.5 |
| FO-MAML | 64.0 | 92.0 | 69.4 | **99.4** | 90.7 | 84.7 | 53.4 | 79.1 | 84.9 | 95.0 | 84.3 | 74.9 | 84.8 | 69.5 | 69.2 | 46.7 | 75.5 | 73.6 | 46.7 | 25.3 | 36.7 | 55.4 |
| i-MAML | 64.6 | 91.5 | 69.5 | **99.4** | 90.9 | 84.4 | 53.8 | 79.2 | **85.8** | 94.6 | 84.6 | 74.9 | 85.0 | 72.2 | 68.8 | 46.4 | 75.5 | **75.3** | 46.7 | 25.6 | 36.6 | 55.9 |
| Reptile | 64.2 | 92.0 | 69.2 | **99.4** | 90.7 | 84.8 | 54.0 | 79.2 | 82.7 | 94.6 | 84.8 | 74.9 | 84.2 | 72.5 | 68.9 | 45.6 | 76.1 | 72.6 | 46.2 | 25.6 | 35.8 | 55.4 |
| ProtoNet | 66.5 | 92.0 | 69.7 | **99.4** | 91.3 | 85.2 | 54.7 | 79.8 | 84.0 | 93.7 | 86.0 | 73.5 | 84.3 | 71.6 | 64.8 | 45.8 | 76.1 | 70.8 | 46.3 | 27.7 | 38.6 | 55.2 |
| ProtoNet-LITE | 64.5 | 91.5 | 68.3 | **99.4** | 91.9 | 84.8 | 53.8 | 79.2 | 81.8 | 94.6 | 86.3 | 74.9 | 84.4 | 71.2 | 64.8 | 42.0 | 78.4 | 69.0 | 46.3 | 27.5 | 42.9 | 55.3 |
| LiFT  $K=1$ | 66.9 | **92.9** | 70.2 | **99.4** | **92.1** | **87.7** | 54.9 | **80.6** | 85.5 | 95.1 | **86.4** | 75.4 | **85.6** | 74.5 | 70.5 | 49.2 | 79.7 | 69.8 | 47.7 | 28.8 | 46.0 | 58.3 |
| $K=2$ | 67.1 | **92.9** | 70.6 | **99.4** | 91.9 | 87.2 | 54.8 | **80.6** | 85.1 | 95.0 | **86.4** | 76.3 | **85.7** | 73.8 | 70.0 | 50.4 | **80.2** | 72.5 | 48.1 | 30.8 | **46.9** | 59.1 |
| $K=3$ | **67.4** | **92.9** | 70.6 | **99.4** | 91.9 | 87.1 | 54.8 | **80.6** | 85.5 | **95.9** | **86.4** | 77.6 | **86.3** | 77.4 | 70.6 | 51.0 | 79.6 | 74.7 | **48.5** | 30.9 | 46.4 | **59.9** |

Table 3: Shakespeare dataset with play characters as tasks. Next word prediction perplexity ($\downarrow$).

| | | Random task split | | Play-wise task split | | Avg |
|---|---|---|---|---|---|---|
| | | tr/va/te=60/10/30 | tr/va/te=80/10/10 | tr/va/te=60/10/30 | tr/va/te=80/10/10 | |
| No Meta Train | | 12.14 | 12.12 | 12.15 | 12.09 | 12.13 |
| Union Train | | 11.30 | 11.22 | 11.35 | 11.19 | 11.27 |
| MAML | | 11.87 | 11.71 | 11.87 | 11.68 | 11.78 |
| FO-MAML | | 11.80 | 11.55 | 11.80 | 11.50 | 11.66 |
| i-MAML | | 12.25 | 12.87 | 12.31 | 12.19 | 12.41 |
| Reptile | | 11.31 | 11.24 | 11.33 | 11.22 | 11.28 |
| LiFT | $K=1$ | **11.01** | **10.97** | **11.08** | 11.00 | **11.02** |
| | $K=2$ | 11.02 | 10.98 | **11.08** | 11.00 | **11.02** |
| | $K=3$ | 11.02 | **10.97** | 11.09 | **10.99** | **11.02** |

the task. We deal with 152 tasks/characters from 36 plays that contain a considerably large amount of text lines, and the goal is to predict the next word given the current and all previous tokens. For the task split, we consider both random split and play-wise split where the latter ensures that no play is overlapped between meta train and test tasks. For each split, we randomly have 60%/10%/30% and 80%/10%/10% train/dev/test task split. We use the BART-base and rank-4 LoRA as the backbone and PEFT adapters. Table 3 shows perplexity scores averaged over test tasks. Overall all methods yield high perplexity implying that next word prediction is a challenging prediction task. However, our LiFT consistently outperforms the multi-task baselines and the meta learning algorithms.

## 4.4 Uncertainty Quantification

As a Bayesian approach, our model can capture uncertainty effectively, and lead to better error calibration than deterministic methods. The error calibration metrics measure how well the prediction accuracy and the prediction confidence are aligned. For instance, the ECE (Guo et al., 2017) is defined as: $ECE = \sum_{b=1}^{B} \frac{N_b}{N} |acc(b) - conf(b)|$ where we partition test instances into $B$ bins along the model's prediction confidence scores, and $conf(b)$, $acc(b)$ are the average confidence and accuracy for the $b$-th bin, respectively. Another metric widely used is the test negative log-likelihood (NLL). For further details, please refer to (Kim & Hospedales, 2023).

On the Cross-fit benchmark, we perform uncertainty quantification on two test tasks for the CLS-45 meta-trained models. Our LiFT is compared to the (deterministic) MAML and two Bayesian meta learning methods: BMAML (Yoon et al., 2018) and ABML (Ravi & Beatson, 2019a). In BMAML, we vary the number of samples $M$ (called particles) in their stochastic ensemble adaptation steps. The results in Table 4 show that our hierarchical Bayesian method leads to considerably better error calibration than previous Bayesian meta learning algorithms on both ECE and NLL metrics.

Table 4: Uncertainty quantification on two test tasks for CLS-45 meta-trained models.

| | | ethos-race | | superglue-cb | |
|---|---|---|---|---|---|
| | | ECE ($\downarrow$) | NLL ($\downarrow$) | ECE ($\downarrow$) | NLL ($\downarrow$) |
| MAML | | 0.1797 | 0.7077 | 0.1679 | 0.7485 |
| | $M=1$ | 0.1727 | 0.6695 | 0.1644 | 0.8406 |
| BMAML | $M=3$ | 0.1507 | 0.6582 | 0.1555 | 0.8069 |
| | $M=5$ | 0.1147 | 0.6423 | 0.1435 | 0.7825 |
| ABML | | 0.1238 | 0.6775 | 0.1572 | 0.9968 |
| | $K=1$ | 0.0900 | 0.3330 | 0.1269 | 0.7118 |
| LiFT (Ours) | $K=3$ | **0.0701** | **0.3021** | 0.1368 | 0.7721 |
| | $K=5$ | 0.0937 | 0.3986 | **0.1186** | **0.6680** |

Table 5: Ablation study on the Cross-Fit benchmark. From our model, we remove either the mixture deviation regularization term at test time (denoted by "Reg (X)"), or the SGLD noise term ("Noise (X)"). Our method modified to handle task-wise pre-fine-tuned models is denoted by "*Fixed $\theta_i^a$s*".

| | CLS-45 | | | CLS-23 | | | CLS-0 | | | MCQA | | | MRC | | | NLI | | | Para. | | |
|---|---|---|---|---|---|---|---|---|---|---|---|---|---|---|---|---|---|---|---|---|---|
| $K$ | 1 | 3 | 5 | 1 | 3 | 5 | 1 | 3 | 5 | 1 | 3 | 5 | 1 | 3 | 5 | 1 | 3 | 5 | 1 | 3 | 5 |
| Reg (X) | 60.1 | 62.6 | 63.4 | 59.9 | 61.5 | 61.9 | 58.7 | 59.6 | 60.1 | 28.7 | 28.9 | 28.9 | 24.7 | 25.0 | 26.1 | 55.5 | 55.6 | 55.6 | 52.6 | 54.4 | 55.9 |
| Noise (X) | 61.0 | 61.8 | 63.6 | 60.9 | 61.9 | 62.7 | 59.4 | 60.3 | 61.0 | 29.1 | 29.1 | 29.4 | 25.5 | 25.7 | 26.8 | 56.0 | 56.2 | 56.5 | 55.5 | 55.8 | 56.9 |
| Fixed $\theta_i^a$s | 61.7 | 62.9 | 63.4 | 61.9 | 63.0 | 62.6 | 61.3 | 61.8 | 61.1 | 28.9 | 29.4 | 29.3 | 26.4 | 28.0 | 28.3 | 56.2 | 55.9 | 56.7 | 55.9 | 56.1 | 56.4 |
| Ours | 63.0 | 63.4 | 64.1 | 60.9 | 62.0 | 63.9 | 60.0 | 61.0 | 61.9 | 29.7 | 30.1 | 30.4 | 27.3 | 27.6 | 28.7 | 56.9 | 56.3 | 56.9 | 56.6 | 56.8 | 57.4 |

## 4.5 ABLATION STUDY

To see the impact of the mixture deviation regularization at test time (i.e., the log-mixture-density term in (18)) in our method, we conduct some ablation study. We also remove the noise term in the SGLD to see the impact of our stochastic method. Table 5 shows that these two components are important for the best performance while the test-time mixture deviation penalty is more crucial.

• **Utilizing Task-wise Pre-fine-tuned Models.** One drawback in our approach is that we require jointly learning all the tasks, which makes it a bit more costly and cumbersome because one has to have all the tasks ready to train jointly, being unable to just use off-the-shelf LoRAs potentially pre-fine-tuned beforehand. However, we can modify our algorithm so as to utilize potentially pre-fine-tuned task-wise LoRAs. The modified scheme we take is as follows: In (7) we replace the Jacobian term $\nabla_\phi \sum_{i=1}^{N} \log p(\theta_i^a | \phi)$ summed over all $\{\theta_i^a\}_{i=1}^{N}$ that are fixed and pre-fine-tuned, by the Jacobian of one single $i$ that is randomly sampled. Thus this can be seen as a noisy stochastic version of the SGLD update for $\phi$. Although one can use the original version (7) in principle, the noisy stochastic version is computationally and practically more attractive without needing to evaluate the Jacobian of the likelihoods over *all* tasks per SGLD update. Our results with this version on Cross-Fit are summarized in Table 5 (titled "*Fixed $\theta_i^a$s*"). For each task split, we pre-fine-tune task-wise LoRAs on meta-train tasks, and fixed them in our method. We still get performance boost in this case, which is promising since we can input pre-fine-tuned LoRA libraries rather than inputting sets of datasets.

## 4.6 EXTRA EXPERIMENTAL RESULTS

Additional ablation studies and extra experimental results including LLMs with larger number of parameters (e.g., FLAN-T5-XL) can be found in Appendix E.

## 5 CONCLUSION

In this paper we have proposed a novel approach to parameter efficient meta fine-tuning, a principled Bayesian method to solve the learning-to-PEFT problem. Our hierarchical Bayesian model, via a reasonable and tractable approximate inference strategy of SGLD-Gibbs and online EM, effectively represents the prior of the LoRA modules that capture the shared information across all training tasks. On a range of NLP and vision cross-task adaptation benchmarks, our model is shown to be superior to not only general meta learning algorithms but also recent LoRA zoo mixing and routing approaches.

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

# Appendix

## Table of Contents

## A  TOY EXPERIMENTS WITH OUR SGLD-GIBBS ALGORITHM

In this section we empirically verify the convergence of the proposed SGLD-Gibbs posterior sampling algorithm. To measure the divergence of the estimated posterior distribution from the true posterior, we devise a true model that admits analytical posterior distribution. Specifically, the following is the true data synthesis process that generates $N$ tasks. We consider the regression problem on the 2D input space.

$$\phi \sim \mathcal{N}(0, \sigma^2 I) \tag{19}$$

$$\theta_i \mid \phi \sim \mathcal{N}(\theta_i; \phi, \beta^2 I) \quad \text{for } i = 1, \ldots, N \tag{20}$$

$$D_i = \{(x_l^i, y_l^i)\}_{l=1}^n \mid \theta_i \sim \prod_{l=1}^n \mathcal{N}(y_l^i; \theta_i^\top x_l^i, \alpha^2) \quad \text{for } i = 1, \ldots, N \tag{21}$$

where $\dim(\phi) = \dim(\theta) = \dim(x) = 2$ and $\dim(y) = 1$. The number of samples per task is set to $n = 20$. We vary the number of tasks $N$ in the experiments below.

From the Gaussian identities, it is not difficult to see that the posterior distribution $p(\phi|\{D_i\}_{i=1}^N)$ can be analytically derived as:

$$p(\phi|\{D_i\}_{i=1}^N) = \mathcal{N}(KY, \sigma^2(I - KX)) \quad \text{where} \tag{22}$$

$$K = X^\top (XX^\top + \sigma^{-2}\Sigma)^{-1} \tag{23}$$

$$X = [X_1^\top, X_2^\top, \cdots, X_N^\top]^\top, \quad X_i = [x_1^i, x_2^i, \cdots, x_n^i]^\top \tag{24}$$

$$Y = [Y_1^\top, Y_2^\top, \cdots, Y_N^\top]^\top, \quad Y_i = [y_1^i, y_2^i, \cdots, y_n^i]^\top \tag{25}$$

$$\Sigma = \mathrm{Diag}(\Sigma_1, \Sigma_2, \cdots, \Sigma_N), \quad \Sigma_i = \alpha^2 I + \beta^2 X_i X_i^\top \tag{26}$$

We compare the original SGLD (6) and our SGLD-Gibbs (9–11). While repeating the recurrences, we collect the posterior samples, and build Gaussian estimates. We vary the number of tasks $N$: 10, 100, 200, and 1000, where $N = 1000$ is larger than those used in all our experiments. The KL divergences of these estimates from the true posterior (22) are shown in Fig. 4. The results indicate that our SGLD-Gibbs converges to the true posterior, and the convergence times are far less than $\times N$ of the original SGLD convergence times.

## B  DETAILED DERIVATIONS FOR OUR ONLINE EM ALGORITHM

In this section we derive our online EM algorithm described in Eq.(13–14) in the main paper. The situation is that the samples $\{\phi^{(m)}\}_m$ are seen in one-by-one manner, that is, $\phi^{(1)} \to \phi^{(2)} \to \cdots$, and we need to update our mixture estimate for each incoming sample. For convenience in derivation, the samples in the incoming order are denoted by $x_1 \to x_2 \to \cdots \to x_{t-1} \to x_t \to \cdots$, and we let $x_t$ be the current sample (instead of $\phi^{(m)}$). Note that in online methods, we are not allowed to store the previous samples $\{x_i\}_{i=1}^{t-1}$.

Before we proceed with deriving our online EM algorithm, we recapitulate the famous and classical *offline* EM algorithm (Dempster et al., 1977) in what follows.

### B.1 RECAP OF OFFLINE EM

By offline, it means that we are allowed to store and access all data samples $\{x_i\}_{i=1}^N$ at once. The Gaussian mixture model is denoted by:

$$p(x; \Theta) = \sum_{j=1}^K \alpha_j \mathcal{N}(x; \mu_j, \Sigma_j) \tag{27}$$

where $\Theta = \{\alpha, \mu, \Sigma\}$ denotes the parameters of the mixture. By Jensen's inequality, it can be shown (e.g., (Dempster et al., 1977)) that the data log-likelihood is bounded below as follows:

$$\sum_{i=1}^N \log p(x_i; \Theta) \geq \sum_{i=1}^N \mathbb{E}_{j \sim q_i} \Big[ \log \big( \alpha_j \mathcal{N}(x_i; \mu_j, \Sigma_j) \big) \Big] + \sum_{i=1}^N H(q_i), \tag{28}$$

which holds for *any* distributions $q_i(j)$ over $j \in \{1, \ldots, K\}$. Here $H(\cdot)$ is the entropy operator. Furthermore, the bound of (28) is tight (i.e., inequality becomes equality) when $q_i$'s are defined as:

$$\text{(E-step)} \quad q_i(j) = \frac{\alpha_j \mathcal{N}(x_i; \mu_j, \Sigma_j)}{\sum_{j'} \alpha_{j'} \mathcal{N}(x_i; \mu_{j'}, \Sigma_{j'})} \tag{29}$$

for all $i = 1, \ldots, N$ and $j = 1, \ldots, K$. Hence this choice of $q_i$s is the optimal choice under given $\Theta$, and the equation (29) constitutes the E-step. One can interpret $q_i(j)$ as the current model $\Theta$'s belief in mixture component assignment: the probability of assigning $x_i$ to the component $j$. With $q_i$'s chosen and fixed as (29), we try to find the maximal value of the right hand side of (28), that is,

$$\max_{\Theta} \sum_{i=1}^N \mathbb{E}_{j \sim q_i} \Big[ \log \big( \alpha_j \mathcal{N}(x_i; \mu_j, \Sigma_j) \big) \Big]. \tag{30}$$

It is easy to see that the optimization (30) admits a closed-form solution (by setting the gradient to 0), which turns out to be (for $j = 1, \ldots, K$):

$$\text{(M-step)} \quad \alpha_j \leftarrow \frac{\sum_{i=1}^N q_i(j)}{\sum_{j'=1}^K \sum_{i=1}^N q_i(j')} \tag{31}$$

$$\mu_j \leftarrow \frac{\sum_{i=1}^N q_i(j) x_i}{\sum_{i=1}^N q_i(j)} \tag{32}$$

$$\Sigma_j \leftarrow S_j - \mu_j \mu_j^\top \quad \text{where } S_j = \frac{\sum_{i=1}^N q_i(j) x_i x_i^\top}{\sum_{i=1}^N q_i(j)} \tag{33}$$

The update equations (31–33) constitute the M-step.

### B.2 DETAILED DERIVATIONS FOR OUR ONLINE EM

Now we consider the online scenario. By online, the samples are observed sequentially $x_1 \rightarrow x_2 \rightarrow \cdots \rightarrow x_{t-1} \rightarrow x_t \rightarrow \cdots$, and at each time step $t$ we only see $x_t$ and it is not allowed to store the previous samples $\{x_i\}_{i=1}^{t-1}$.

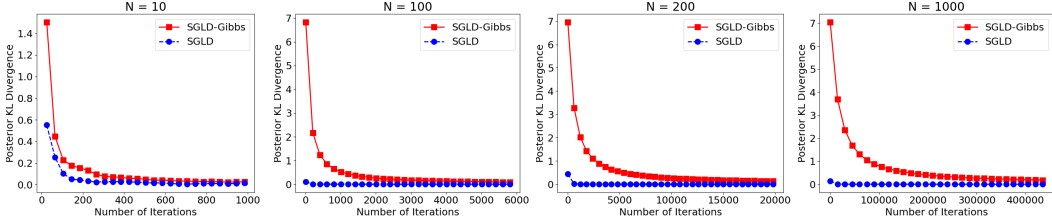

Figure 4: (Toy experiment for SGLD-Gibbs) KL divergences between the true posterior and the estimated posteriors (SGLD vs. SGLD-Gibbs) measured at several SGLD-Gibbs iteration numbers. Since we use the number of SGLD-Gibbs iterations for the X-axis, we match the result of the original SGLD by extrapolating its number of iterations (roughly, the original SGLD learning curve can be retrieved by taking the first one $N$-th of the X-axis).

We denote our mixture model right before seeing $x_t$ by:

$$p(x; \Theta_{t-1}) = \sum_{j=1}^{K} \alpha_j^{t-1} \mathcal{N}(x; \mu_j^{t-1}, \Sigma_j^{t-1}) \tag{34}$$

where $\Theta_{t-1} = \{\alpha^{t-1}, \mu^{t-1}, \Sigma^{t-1}\}$ is our estimate of the mixture just before seeing $x_t$.

If we pretend that we can access all the previous samples $\{x_i\}_{i=1}^{t-1}$ together with the current one $x_t$, then following the offline E-step (29), our (fictitious) E-step would be as follows:

$$q_i(j) = \frac{\alpha_j^{t-1} \mathcal{N}(x_i; \mu_j^{t-1}, \Sigma_j^{t-1})}{\sum_{j'} \alpha_{j'}^{t-1} \mathcal{N}(x_i; \mu_{j'}^{t-1}, \Sigma_{j'}^{t-1})} \quad \text{for } i = 1, \ldots, t. \tag{35}$$

Now we define the following quantities $(n_j)$ as the current model $\Theta_{t-1}$'s component assignment accumulation for previous samples $\{x_i\}_{i=1}^{t-1}$:

$$n_j := \sum_{i=1}^{t-1} q_i(j) \quad \text{for } j = 1, \ldots, K. \tag{36}$$

We want to emphasize here that although $\{q_i(j)\}_{i=1}^{t-1}$ would not be available in the online scenario, the quantities $\{n_j\}_{j=1}^{K}$ succinctly capture the key information about those previous samples, which is all we need during the M-step. More specifically, we can re-write the M-step in (31–33) as follows:

$$\alpha_j^t \leftarrow \frac{\sum_{i=1}^{t} q_i(j)}{\sum_{j'=1}^{K} \sum_{i=1}^{t} q_i(j')} = \frac{n_j + q_t(j)}{\sum_{j'=1}^{K} \left(n_{j'} + q_t(j')\right)} = \frac{n_j + q_t(j)}{1 + \sum_{j'=1}^{K} n_{j'}} \tag{37}$$

$$\mu_j^t \leftarrow \frac{\sum_{i=1}^{t} q_i(j)x_i}{\sum_{i=1}^{t} q_i(j)} = \frac{\sum_{i=1}^{t-1} q_i(j)x_i + q_t(j)x_t}{n_j + q_t(j)} \approx \frac{n_j \mu_j^{t-1} + q_t(j)x_t}{n_j + q_t(j)} \tag{38}$$

$$S_j^t \leftarrow \frac{\sum_{i=1}^{N} q_i(j)x_i x_i^\top}{\sum_{i=1}^{N} q_i(j)} = \frac{\sum_{i=1}^{t-1} q_i(j)x_i x_i^\top + q_t(j)x_t x_t^\top}{n_j + q_t(j)} \approx \frac{n_j S_j^{t-1} + q_t(j)x_t x_t^\top}{n_j + q_t(j)} \tag{39}$$

Note that the approximations in (38) and (39) originate from the M-step for $\mu$ and $S$ at the previous time $t-1$, respectively, that is,

$$\mu_j^{t-1} \leftarrow \frac{\sum_{i=1}^{t-1} q_i(j)x_i}{\sum_{i=1}^{t-1} q_i(j)}, \qquad S_j^{t-1} \leftarrow \frac{\sum_{i=1}^{t-1} q_i(j)x_i x_i^\top}{\sum_{i=1}^{t-1} q_i(j)} \tag{40}$$

What remains is to estimate $\{n_j\}_{j=1}^{K}$ in an online fashion without explicitly computing (36) using the previous samples $\{x_i\}_{i=1}^{t-1}$. Assuming that we have a good estimate of $n_j$ at time $t-1$, we can update $n_j$ for time $t$ by: $n_j \leftarrow n_j + q_t(j)$ where $q_t(j)$ would be computed similarly as (35) using $\Theta_t$. Although this can be done with the updated $\Theta_t$ after the M-step (37–39), we can reuse the current $q_t(j)$ to have an approximate estimate, which rarely incurs significant difference in practice.

In summary, at each time step $t$, we only need to compute $\{q_t(j)\}_{j=1}^{K}$ for the current sample $x_t$ (E-step), and the M-step (37–39) can be done without accessing the previous samples $\{x_i\}_{i=1}^{t-1}$. This completes the derivation for our online EM algorithm.

## C  EXPERIMENTAL DETAILS

All experiments are conducted on a single V100 GPU. For the scale hyperparameters in our LiFT model, we always use $\sigma = 0.01$ (scale of prior $p(\phi)$) and $\beta = 0.01$ (scale of $p(\theta_i|\phi)$). For the pre-trained backbone, we use the pre-trained BART-base checkpoint from HuggingFace for Cross-Fit and Shakespeare datasets, while the ImageNet-pre-trained ViT-B/16 for the VTAB benchmark. The SGLD-Gibbs algorithm takes the burn-in steps 2000 with warmup 1000 steps for the NLP datasets and 200 burn-in and 100 warmup for the vision. The learning rates are $10^{-3}$ for the NLP and $10^{-4}$ for the VTAB. We use the batch size 16 with 20K steps for the NLP tasks while we take batch size 128 and 10K steps for the vision task. For the meta learning baselines (e.g., MAML, its variants and Reptile), we use similar hyperparameters, but the learning rates are adjusted for numerical stability. The inner loop learning rates are typically chosen 5 times the outer learning rate.

CLS-45

"train": ["ade_corpus_v2-classification", "circa", "ethos-directed_vs_generalized", "ethos-disability", "ethos-gender",
        "ethos-sexual_orientation", "glue-cola", "glue-mnli", "glue-mrpc", "glue-qnli", "glue-qqp", "glue-rte",
        "glue-sst2", "glue-wnli", "google_wellformed_query", "hate_speech_offensive", "hatexplain", "imdb",
        "kilt_fever", "liar", "onestop_english", "paws", "rotten_tomatoes", "scicite", "scitail", "sick", "sms_spam",
        "superglue-rte", "superglue-wic", "superglue-wsc", "trec", "trec-finegrained", "tweet_eval-emoji",
        "tweet_eval-emotion", "tweet_eval-irony", "tweet_eval-offensive", "tweet_eval-sentiment",
        "tweet_eval-stance_abortion", "tweet_eval-stance_climate", "tweet_eval-stance_hillary"]

"dev": ["ag_news", "climate_fever", "ethos-national_origin", "hate_speech18", "medical_questions_pairs",
        "poem_sentiment", "tweet_eval-hate", "tweet_eval-stance_atheism", "tweet_eval-stance_feminist"]

"test": ["anli", "ethos-race", "ethos-religion", "superglue-cb", "tab_fact", "wiki_qa", "yelp_polarity"]

CLS-30

"train": ["ade_corpus_v2-dosage", "blimp-ellipsis_n_bar_2", "blimp-sentential_negation_npi_scope",
        "commonsense_qa", "crows_pairs", "duorc", "ethos-disability", "ethos-sexual_orientation", "glue-cola",
        "glue-mnli", "glue-mrpc", "glue-qqp", "glue-rte", "glue-wnli", "hatexplain", "hellaswag", "imdb",
        "kilt_zsre", "lama-google_re", "lama-squad", "math_qa", "numer_sense", "openbookqa", "paws", "piqa",
        "proto_qa", "quartz-no_knowledge", "race-high", "ropes", "scicite", "sciq", "sick", "sms_spam",
        "superglue-rte", "superglue-wsc", "tweet_eval-emotion", "tweet_eval-offensive", "tweet_eval-sentiment",
        "tweet_eval-stance_hillary"]

"dev": ["ag_news", "climate_fever", "ethos-national_origin", "hate_speech18", "medical_questions_pairs",
        "poem_sentiment", "tweet_eval-hate", "tweet_eval-stance_atheism", "tweet_eval-stance_feminist"]

"test": ["anli", "ethos-race", "ethos-religion", "superglue-cb", "tab_fact", "wiki_qa", "yelp_polarity"]

CLS-0

"train": ["ade_corpus_v2-dosage", "art", "biomrc", "blimp-anaphor_number_agreement", "blimp-ellipsis_n_bar_2",
        "blimp-sentential_negation_npi_licensor_present", "blimp-sentential_negation_npi_scope",
        "break-QDMR-high-level", "commonsense_qa", "crows_pairs", "dream", "duorc", "eli5-asks", "eli5-eli5",
        "freebase_qa", "gigaword", "hellaswag", "hotpot_qa", "kilt_ay2", "kilt_hotpotqa", "kilt_trex", "kilt_zsre",
        "lama-conceptnet", "lama-google_re", "lama-squad", "math_qa", "numer_sense", "openbookqa", "piqa",
        "proto_qa", "qa_srl", "quarel", "quartz-no_knowledge", "race-high", "reddit_tifu-title", "reddit_tifu-tldr",
        "ropes", "sciq", "social_i_qa", "spider", "superglue-multirc", "wiki_bio", "wikisql", "xsum",
        "yelp_review_full"]

  "dev": ["tweet_eval-stance_feminist", "ethos-national_origin", "tweet_eval-hate", "ag_news",
        "amazon_polarity", "hate_speech18", "poem_sentiment", "climate_fever", "medical_questions_pairs",
        "tweet_eval-stance_atheism"]

  "test": ["superglue-cb", "dbpedia_14", "wiki_qa", "emo", "yelp_polarity", "ethos-religion",
        "financial_phrasebank", "tab_fact", "anli", "ethos-race"]

MCQA

"train": ["adversarialqa", "boolq", "duorc", "eli5-askh", "eli5-asks", "eli5-eli5", "freebase_qa", "hotpot_qa",
        "kilt_hotpotqa", "kilt_nq", "kilt_trex", "kilt_zsre", "lama-conceptnet", "lama-google_re", "lama-squad",
        "lama-trex", "mc_taco", "numer_sense", "quoref", "ropes", "squad-no_context", "squad-with_context",
        "superglue-multirc", "superglue-record", "web_questions"],

"dev": ["codah", "dream", "qasc", "race-high"],

"test": ["ai2_arc", "aqua_rat", "commonsense_qa", "cosmos_qa", "hellaswag", "math_qa", "openbookqa", "quail",
        "quarel", "quartz-no_knowledge", "quartz-with_knowledge", "race-middle", "sciq", "social_i_qa",
        "superglue-copa", "swag", "wino_grande", "wiqa"]

Figure 5: Cross-Fit 7 task splits (Part I).

## D    EXTENDED RELATED WORK

**General multi-task meta learning.** The gradient-based adaptation technique MAML (Finn et al., 2017b) is recognized as one of the most popular episodic meta learning algorithms due to its empirical success and known theoretical support (Chen & Chen, 2022). Several variants have been proposed to

```
┌ MRC ────────────────────────────────────────────────────────────────────────────┐
│                                                                                   │
│ "train": ["ai2_arc", "aqua_rat", "boolq", "codah", "commonsense_qa", "cosmos_qa", "dream", "eli5-askh", │
│         "eli5-asks", "eli5-eli5", "freebase_qa", "hellaswag", "kilt_hotpotqa", "kilt_nq", "kilt_trex", "kilt_zsre", │
│         "lama-conceptnet", "lama-google_re", "lama-squad", "lama-trex", "math_qa", "mc_taco", "numer_sense", │
│         "openbookqa", "qasc", "quail", "quarel", "quartz-no_knowledge", "quartz-with_knowledge", "race-high", │
│         "race-middle", "sciq", "social_i_qa", "squad-no_context", "superglue-copa", "superglue-multirc", "swag", │
│         "web_questions", "wino_grande", "wiqa"],                                   │
│                                                                                   │
│ "dev": ["duorc", "squad-with_context"],                                           │
│                                                                                   │
│ "test": ["adversarialqa", "hotpot_qa", "quoref", "ropes", "superglue-record"]     │
└───────────────────────────────────────────────────────────────────────────────────┘
┌ NLI ─────────────────────────────────────────────────────────────────────────────┐
│                                                                                   │
│ "train": ["ade_corpus_v2-classification", "ag_news", "circa", "climate_fever", "ethos-directed_vs_generalized", │
│         "ethos-disability", "ethos-gender", "ethos-national_origin", "ethos-race", "ethos-religion", │
│         "ethos-sexual_orientation", "glue-cola", "glue-mrpc", "glue-qqp", "glue-sst2", "google_wellformed_query", │
│         "hate_speech18", "hate_speech_offensive", "hatexplain", "imdb", "kilt_fever", "liar", │
│         "medical_questions_pairs", "onestop_english", "paws", "poem_sentiment", "rotten_tomatoes", "scicite", │
│         "sick", "sms_spam", "superglue-wic", "superglue-wsc", "tab_fact", "trec", "trec-finegrained", │
│         "tweet_eval-emoji", "tweet_eval-emotion", "tweet_eval-hate", "tweet_eval-irony", "tweet_eval-offensive", │
│         "tweet_eval-sentiment", "tweet_eval-stance_abortion", "tweet_eval-stance_atheism", │
│         "tweet_eval-stance_climate", "tweet_eval-stance_feminist", "tweet_eval-stance_hillary", "wiki_qa", │
│         "yelp_polarity"],                                                          │
│                                                                                   │
│ "dev": ["art", "glue-mnli", "superglue-rte"],                                     │
│                                                                                   │
│ "test": ["anli", "glue-qnli", "glue-rte", "glue-wnli", "scitail", "sick", "superglue-cb"] │
└───────────────────────────────────────────────────────────────────────────────────┘
┌ Paraphrasing ────────────────────────────────────────────────────────────────────┐
│                                                                                   │
│ "train": ["ade_corpus_v2-classification", "ag_news", "anli", "circa", "climate_fever", │
│         "ethos-directed_vs_generalized", "ethos-disability", "ethos-gender", "ethos-national_origin", │
│         "ethos-race", "ethos-religion", "ethos-sexual_orientation", "glue-cola", "glue-mnli", "glue-qnli", │
│         "glue-rte", "glue-sst2", "glue-wnli", "google_wellformed_query", "hate_speech18", │
│         "hate_speech_offensive", "hatexplain", "imdb", "kilt_fever", "liar", "onestop_english", │
│         "poem_sentiment", "rotten_tomatoes", "scicite", "scitail", "sick", "sms_spam", "superglue-cb", │
│         "superglue-rte", "superglue-wic", "superglue-wsc", "tab_fact", "trec", "trec-finegrained", │
│         "tweet_eval-emoji", "tweet_eval-emotion", "tweet_eval-hate", "tweet_eval-irony", "tweet_eval-offensive", │
│         "tweet_eval-sentiment", "tweet_eval-stance_abortion", "tweet_eval-stance_atheism", │
│         "tweet_eval-stance_climate", "tweet_eval-stance_feminist", "tweet_eval-stance_hillary", "wiki_qa", │
│         "yelp_polarity"],                                                          │
│                                                                                   │
│ "dev": ["glue-qqp"],                                                              │
│                                                                                   │
│ "test": ["glue-mrpc", "medical_questions_pairs", "paws"]                          │
└───────────────────────────────────────────────────────────────────────────────────┘
```

Figure 6: Cross-Fit 7 task splits (Part II).

enhance MAML's performance as well as addressing its computational overhead, including the L2 regularization technique (Rajeswaran et al., 2019), the first-order approximation (Nichol et al., 2018), and probabilistic extension (Finn et al., 2018). Bayesian extensions to MAML (Finn et al., 2018; Yoon et al., 2018; Ravi & Beatson, 2019b; Nguyen et al., 2020) are highly related to our work, however, they are weak in several aspects. For instance, BMAML (Yoon et al., 2018) is not a hierarchical Bayesian model, but turns MAML's gradient-based *deterministic* adaptation steps to a *stochastic* counterpart using the particle-based stochastic ensemble-based adaptation steps. ABML (Ravi & Beatson, 2019a) relies on the variational inference, and resorts to a Gaussian posterior approximation.

**PEFT/LoRA adapter mixture/ensemble techniques.** Especially in the NLP domain with emphasis on cross-task adaptation (Ye et al., 2021), a large number of techniques have been proposed. Considering the large scale of the backbone networks (e.g., LLMs), many recent approaches aim to focus on adapting relatively smaller PEFT modules instead of the full backbone adaptation as in Ye et al. (2021). As it is overwhelming to carry out all of the literature reviews in this paper, we just highlight here a few works that are the most relevant to our method. Meta-ICL (Min et al., 2022) aims to mimic the test scenario where the few-shot context data are fed into the prompt in the LLM

inference. This motivates to adapt a model with the same type of few-shot prompting during the meta training. However, a main drawback is computational overhead in dealing with potentially large downstream training data at both training and test time. Several recent approaches aim to build a model (LoRA) zoo of task-wise trained adapters (LoRAs) for multi-task training. The model zoo undergoes certain mixing of merging steps so as to help optimal model selection at test time. The Mixture-of-LoRAs (Feng et al., 2024; Asadi et al., 2024) use the router (MoE), an MLP network, that takes all task-wisely trained LoRAs, which makes it computationally infeasible as the number of tasks increases. Alternatively the mixture composition can be learned from data as in (Asadi et al., 2024), however, it also suffers from the same computational issue for a large number of tasks. LoRA Retriever (Zhao et al., 2024a) selects the most relevant LoRAs from the model zoo using the cosine similarity, and combine the selected LoRAs by either mixture (weight matrix-level mixing) or fusion (low-rank matrix-level mixing) operations. Another recent strategy is the model-based clustering (Ostapenko et al., 2024), where they perform the k-means clustering to compute the cluster assignment of the tasks. With this task assignment, they re-train LoRAs cluster-wisely. Although there are several proposed mixing strategies such as the instance-wise arrow routing, they show that the performance of the simple LoRA averaging performs comparably well.

**Modulation-based meta learning.** Recently there have been several works that consider meta-learning by updating relatively few parameters (modulation). CAVIA (Zintgraf et al., 2019) is the first paper that suggested meta-learning with modulation. CNAPS (Requeima et al., 2019) uses a set-based architecture that takes the entire support set and the query image as input, and outputs a query label. The FiLM (Perez et al., 2018) modulation has been combined with MAML (Finn et al., 2017b) in (Dupont et al., 2022a;b) while LoRA (Hu et al., 2022) modulation has been applied to MAML in (Schwarz et al., 2023). LITE (Bronskill et al., 2021) aims to build a meta learner using the entire support data instead of a minibatch, but to remedy computational overhead they set a randomly chosen large portion of the support data as non-backpropagatable. In (Bateni et al., 2020) a simple class-covariance-based distance metric has been adopted into CNAPS. In (Tack et al., 2024), a feature extraction and memory-augmentation approach has been proposed to compress and extract information from new data into compact modulations stored in a memory bank. It has a very different flavor from ours in that they do not optimize PEFT parameters but to predict them based on query. Unlike most of these modulation approaches, our approach is network architecture independent, and can potentially work smoothly with any form of backbone networks.

**Online EM algorithms.** There have been several prior works on the online EM methods in the literature. Although we find that none of them are identical to the one proposed in this paper, most are similar in nature to ours. In (Neal & Hinton, 1998) they proposed an incremental version of the EM (iEM), in which at each iteration the conditional expectation of the latent variable (E-step) is updated only for a mini-batch of the observations. In (Cappé & Moulines, 2009) the full-batch E-step is replaced by a stochastic approximation step, closely related to the stochastic gradient step. This scheme is further extended in (Chen et al., 2018) for variance reduction. In (Karimi et al., 2019) they analyzed incremental and stochastic versions of the EM algorithm as well as extending the variance reduction technique in a common unifying framework. The online EM algorithms have been applied to the graph learning problem (He & Wai, 2022), the filtering problem in state-space models (Özkan et al., 2012), and Internet traffic modeling (Liu et al., 2006). In the latter, they deal with Poisson mixtures where the online algorithm is based on the fact that the parameter increment has a positive projection on the gradient of the likelihood function. They formed a stochastic approximation update scheme via exponential averaging of the old mixture parameters and new one.

# E    EXTRA EXPERIMENTAL RESULTS

## E.1    RESULTS ON FLAN-T5-XL

To verify that the proposed approach is scalable to even larger LLMs, we ran our model on FLAN-T5-XL (Chung et al., 2022) which contains 3B parameters. The results on the Crossfit CLS-45 task split are summarized in Table 6 (compared with BART-base results in the main paper). We ran all methods on a single A100 80GB GPU where i-MAML (and its mixtures) and BMAML baselines incurred the out-of-memory issue. Our LiFT model runs well, and as shown we have large improvements over the competing methods.

Table 6: FLAN-T5-XL (3B) results on the Crossfit CLS-45 task split.

|  | No Meta Train | Union Train | MAML | FO-MAML | Reptile | LiFT $K=1$ | LiFT $K=3$ | LiFT $K=5$ |
|---|---|---|---|---|---|---|---|---|
| FLAN-T5-XL | 65.13 | 73.41 | 73.74 | 74.22 | 69.39 | 78.37 | 79.69 | **80.01** |
| BART-base | 57.34 | 57.25 | 52.68 | 59.38 | 57.59 | 62.96 | 63.37 | **64.12** |

Table 7: Comparison against non-hierarchical Bayesian models on the Crossfit CLS-45 task split.

|  | Non-Hierarchical | | Hierarchical | | |
|---|---|---|---|---|---|
|  | Shared PEFT | Task-wise PEFT | LiFT $K=1$ | LiFT $K=3$ | LiFT $K=5$ |
| CLS-45 | 55.62 | 56.74 | 62.96 | 63.37 | **64.12** |

Table 8: Ablation study on the prior variance hyperparameters $(\sigma^2, \beta^2)$ for our LiFT $K=5$.

| $(\sigma^2, \beta^2)$ | 0.001 | 0.005 | 0.01 | 0.05 | 0.1 | 0.2 | 0.3 | 0.5 |
|---|---|---|---|---|---|---|---|---|
| CLS-45 | 62.87 | 63.05 | 64.12 | 63.62 | 62.75 | 62.75 | 61.48 | 61.73 |

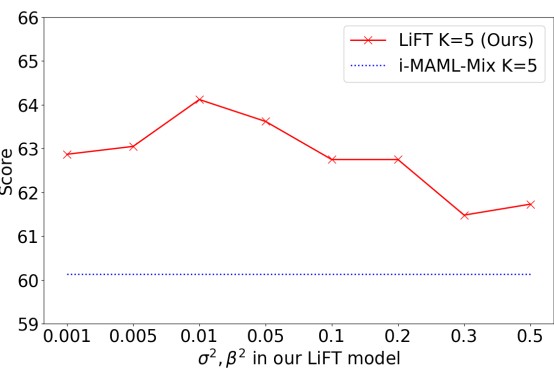

Figure 7: Ablation study on the prior variance hyperparameters $(\sigma^2, \beta^2)$ for our LiFT $K = 5$. Contrasted with the best runner-up baseline i-MAML-Mix $K = 5$.

### E.2 IMPACT OF HIERARCHICAL BAYESIAN MODELING

To show the importance of hierarchical modeling, we have implemented and tested two *non-hierarchical* Bayesian models. In the first model, the PEFT LoRA parameters are treated as random variables as ours, but they are shared across all tasks. Hence the posterior inference in this case amounts to learning task-shared (task-agnostic) information from meta training data. We used the SGLD posterior inference. This model performs way worse than our hierarchical model as shown in Table 7.

As a second non-hierarchical Bayesian model, we can think of a model where each task is represented by its own PEFT LoRA parameters, but there is no governing higher-level random variables as in our LiFT model. Consequently this model aims to learn only task-specific information, and would not transfer anything to novel tasks. Hence it can be seen as a Bayesian version of "No Meta Train" method, which only runs test-time adaptation via SGLD posterior inference. This model also performs poorly compared to our hierarchical model as shown in Table 7.

### E.3 ABLATION STUDY ON HYPERPARAMETERS $\sigma^2$ AND $\beta^2$

Recall that we have variance hyperparameters $(\sigma^2, \beta^2)$ for the priors $p(\phi)$ and $p(\theta_i^a|\phi)$ in (2). To see the robustness to these hyperparameters, we have run the ablation study with our LiFT $K = 5$ model on the Crossfit CLS-45 task split. As shown in Table 8 and Fig. 7, the test performance is quite robust to the choice of $(\sigma^2, \beta^2)$.

Table 9: Ablation study on our asynchronous update strategy Eq. (9–11) vs. stochastic approximation alternative. Both strategies use LiFT $K = 5$.

|        | Stochastic approximation | Proposed asynchronous $J$ update |
|--------|--------------------------|----------------------------------|
| CLS-45 | 63.61                    | **64.12**                        |

Table 10: Per-iteration wall clock training times on the Crossfit CLS-45 task split. For our LiFT $K = 5$, this is the per-task/iteration time after the burn-in stage.

|                  | Rank-4 LoRA  | Rank-64 LoRA |
|------------------|--------------|--------------|
| Our LiFT $K = 5$ | 0.268 secs   | 0.287 secs   |
| Union Train      | 0.242 secs   | 0.270 secs   |
| MAML             | 0.372 secs   | 0.380 secs   |

Table 11: Warm-up and burn-in stage running times on the Crossfit CLS-45 task split.

|               | Rank-4 LoRA | Rank-64 LoRA |
|---------------|-------------|--------------|
| Warm-up time  | 219 secs    | 233 secs     |
| Burn-in time  | 277 secs    | 279 secs     |

### E.4    Ablation Study on Asynchronous Update Strategy Eq. (9–11)

Apart from our asynchronous $\phi$ update strategy with the auxiliary variable $J$ in Eq. (9–11), one can think of a stochastic approximate version of (7), which essentially replaces the average of $\nabla_\phi \log p(\theta_i|\phi)$ over tasks $i$ by the current task iterate alone. To compare this simpler strategy with our asynchronous $J$ update strategy, we have run the stochastic approximate version on the Crossfit CLS-45 task split. As shown in Table 9, the simple stochastic approximation has decent performance, but slightly underperforms our original proposal of the $J$ update strategy.

### E.5    Wall Clock Running Times

We report the wall clock training times (on a single V100 GPU) in Table 10 and 11. Our LiFT with $K = 5$ GMM is compared with the two baselines "Union Train" and "MAML" on two different PEFT (rank 4 and 64) cases. The result in Table 10 signifies that our model is as efficient as the vanilla SGD update (i.e., "Union Train") for both low and high ranks, thanks to our efficient SGLD-Gibbs sampler. Asymptotically (in Big-$O$ notation), both the vanilla SGD update and our SGLD-Gibbs (Eq. (9)–(11)) + online EM (Eq. (13)–(14)) take the same $O(T + d)$ time per iteration where $T$ is the time for the backbone forward/backward computation for evaluating $\nabla_{\theta_i^a} \log p(D_i|\theta_i^a)$ and $d$ is the number of PEFT parameters. This is because of the Gaussian $p(\theta_i^a|\phi)$ which allows analytic gradient $\nabla \log p(\theta_i^a|\phi)$ with respect to both $\theta_i^a$ and $\phi$. The EM iteration also takes $O(d)$ time.

The warm-up steps in our method refers to the vanilla SGD steps for the first 1000 iterations, and the burn-in steps amounts to running the proposed SGLD recurrences for the next 1000 iterations (without collecting the posterior samples). After the burn-in steps, we collect the posterior samples while running the SGLD. The wall clock times for these two stages are shown in Table 11, and they are quite reasonable times.

### E.6    Comparison with Bootstrapped-MAML (Flennerhag et al., 2022)

For empirical comparison with more recent meta learning approaches, we have implemented the Bootstrapped-MAML (Flennerhag et al., 2022) in our Crossfit experimental framework with LoRA PEFT as parameters to learn. In Bootstrapped-MAML, the target model is first obtained by applying SGD updates on the train data then batch data sequentially. The meta learning loss is then the KL divergence between the predictive distributions of the train-data-updated model and the target model,

Table 12: Comparison with Bootstrapped-MAML on the Crossfit CLS-45 task split.

|        | MAML  | Bootstrapped-MAML | LiFT $K=1$ | LiFT $K=3$ | LiFT $K=5$ |
|--------|-------|-------------------|------------|------------|------------|
| CLS-45 | 52.68 | 55.02             | 62.96      | 63.37      | **64.12**  |

Table 13: Comparison with MAML with multiple inner iterations.

| # Inner iterations in MAML | 1     | 2     | 3     | 5     | LiFT $K=5$ |
|----------------------------|-------|-------|-------|-------|------------|
| CLS-45                     | 52.68 | 48.48 | 55.73 | 56.02 | **64.12**  |

Table 14: Statistical significance and standard deviations. The $p$-values of the existing methods against our LiFT-$K=5$ are shown in the parentheses.

|        | No Meta Train | Union Train | MAML | FO-MAML | i-MAML | Reptile |
|--------|---------------|-------------|------|---------|--------|---------|
| CLS-45 | $56.27 \pm 1.30$ $(3.26 \times 10^{-6})$ | $56.79 \pm 0.59$ $(3.63 \times 10^{-8})$ | $52.82 \pm 1.05$ $(3.72 \times 10^{-8})$ | $57.70 \pm 1.18$ $(8.02 \times 10^{-6})$ | $59.03 \pm 0.65$ $(8.81 \times 10^{-7})$ | $58.26 \pm 0.66$ $(3.36 \times 10^{-7})$ |

|        | BMAML M=5 | ABML | MAML-Mix K=5 | i-MAML-Mix K=5 | Retriever Mix K=5 | Retriever Mix K=all |
|--------|-----------|------|--------------|----------------|-------------------|---------------------|
| CLS-45 | $57.98 \pm 0.59$ $(1.09 \times 10^{-7})$ | $55.74 \pm 1.25$ $(1.47 \times 10^{-6})$ | $53.87 \pm 0.12$ $(7.19 \times 10^{-12})$ | $58.99 \pm 1.20$ $(4.85 \times 10^{-5})$ | $55.82 \pm 1.42$ $(4.06 \times 10^{-6})$ | $52.58 \pm 1.11$ $(4.87 \times 10^{-8})$ |

|        | Retriever Fuse K=5 | Retriever Fuse K=all | MBC-$\mu$ K=5 | MBC-$\mu$ K=10 | LiFT K=5 (Ours) | |
|--------|--------------------|----------------------|---------------|----------------|-----------------|---|
| CLS-45 | $56.03 \pm 1.69$ $(1.69 \times 10^{-5})$ | $53.77 \pm 1.29$ $(3.52 \times 10^{-7})$ | $56.17 \pm 1.05$ $(6.23 \times 10^{-7})$ | $55.41 \pm 0.87$ $(8.08 \times 10^{-8})$ | $63.87 \pm 0.31$ | |

where the latter is treated as constant (no backpropagation). The comparison with Bootstrapped-MAML on the Crossfit CLS-45 split is shown in Table 12.

### E.7 COMPARISON WITH MAML (FINN ET AL., 2017B) WITH MULTIPLE INNER ITERATIONS

We have run the experiments to see the effect of the multiple inner loop updates in MAML. On the Cross-fit CLS-45 task split, increasing the number of inner iterations often helps improve the test performance, but MAML still underperforms our LiFT by large margin (Table 13). More than 5 iterations in MAML incurred the out-of-memory issues on a V100 GPU.

### E.8 STATISTICAL SIGNIFICANCE

Due to the computational overhead, it is time- and compute-expensive to run multiple random seed experiments. However, to measure the statistical significance, we perform multiple 5 random seed runs on the Crossfit-CLS45 split to collect standard deviations and compute $p$-values. As shown in Table 14, our LiFT model outperforms the existing methods statistically significantly.

