# OpenReview forum: "LiFT: Learning to Fine-Tune via Bayesian Parameter Efficient Meta Fine-Tuning"
_ICLR.cc/2025/Conference — ICLR 2025 Spotlight_

### Official Review · Reviewer_pSzU · 2024-10-24

**Soundness:** 3
**Presentation:** 4
**Contribution:** 3
**Rating:** 8
**Confidence:** 3

**Summary:**

This paper presents a novel hierarchical Bayesian model to tackle the cross-task PEFT knowledge transfer problem with meta-learning. The proposed LiFT approach introduces a SGLD-Gibbs sampling algorithm for efficient training and an online EM algorithm to maintain a Gaussian mixture representation of the posterior samples in an online manner. This method is evaluated on both NLP and vision tasks, and the results show improved performance over several baselines.

**Strengths:**

*  The paper tackles the cross-task PEFT knowledge transfer problem, which is a relevant and important topic in meta-learning and fine-tuning large models.
*  The paper is well written and well structured. The design concept is clear from the very beginning. The reader is guided through the method step-by-step and the theoretical foundations are supported by the experiments.
*  The integration of Gibbs sampling into SGLD and the assumptions made to approximate the posterior (all the variables are visited a sufficient number of times and frequently) are valuable.
*  The proposed approach can also be applied when pre-fine-tuned models are already available, which is valuable for knowledge reuse.

**Weaknesses:**

*  The meta-learning baselines considered in the experiments are limited. I recommend including more recent deterministic and bayesian meta-learning approaches. Some examples, but not restriced to these, are SNAIL [2], ProtoNet [3], and MetaQDA [4].
*  The results do not include standard deviations, making it difficult to assess whether the improvements with LiFT are statistically significant.
*  While the work addresses several challenges, it also has some limitations, such as the joint training on the training tasks, which can be infeasible for large-scale datasets. The authors might consider adding a limitations section to acknowledge this constraint and any other potential challenges the method faces.
*  It is unclear which architectures and hyperparameters are used for the meta-learning baselines. I assume that, for a fair comparison, ViT-B/16 was employed. However, MAML is generally inefficient when adapted to large feature extractors due to the difficulty of finding an optimal meta-initialization, which scales with the overall parameter space.
*  A single inner step is used for MAML and its variant. While this reduces the computational cost, it is unclear if these results reflect the best possible performance for MAML, As noted in [1], MAML typically benefits from multiple inner loop updates.

References: \
[1] Han-Jia Ye, & Wei-Lun Chao (2022). How to Train Your MAML to Excel in Few-Shot Classification. In International Conference on Learning Representations. \
[2] Mishra, N., Rohaninejad, M., Chen, X., & Abbeel, P. (2017). A simple neural attentive meta-learner. arXiv preprint arXiv:1707.03141.\
[3] Snell, J., Swersky, K., & Zemel, R. (2017). Prototypical networks for few-shot learning. Advances in neural information processing systems, 30.\
[4] Zhang, X., Meng, D., Gouk, H., & Hospedales, T. (2021). Shallow bayesian meta learning for real-world few-shot recognition. In Proceedings of the IEEE/CVF international conference on computer vision (pp. 651–660).\

**Questions:**

*  How does the choice of $\sigma$ and $\beta$ impact on the model performance?
*  In Figure 4 and the related discussion in line 742, it does seem that SGLD converges faster than SGLD-Gibbs especially for a low number of iterations. Could the authors clarify how to interpret this figure?
*  Could the choice of parameters selected for meta-learning in the MAML-based baselines be explained in more detail?
*  Could a comparison with MAML using 5 inner loops be provided to ensure that performance is not affected by the choice of a single inner loop? If conducting this experiment is not feasible due to time constraints, could an estimate of the time and computational resources required to complete it (based on the time taken for 1 inner loop) be provided?

---

> ### Author Response · Authors · 2024-11-21
> **Thank you for your valuable feedback!**
>
> > 1. More meta-learning baselines.
>
> Most methods the reviewer suggested assume small class cardinality cases (eg, learning the class prototypes or linear readout heads), hence they are not straightforwardly applicable to open-ended NLP (Crossfit) benchmarks. On the VTAB benchmark, however, we have been able to implement ProtoNet and more recent memory-efficient version ProtoNet-LITE. The latter (LITE) essentially aims to represent the prototype vectors using the entire support data (instead of a minibatch), but to remedy computational overhead they set a randomly chosen large portion of the support data as non-backpropagatable.
>
> [ProtoNet-LITE] Memory Efficient Meta-Learning with Large Images, NeurIPS 2021
>
> The results are shown in the following table (the full scores for individual tasks are shown in Table 2 of the revised paper).
>
> |  | Non-natural -> Natural | Non-special -> Special | Non-structured -> Structured |
> |:----:|:----:|:----:|:----:|
> | No Meta Train | 78.0 | 84.3 | 54.4 |
> | Union Train | 79.0 | 84.5 | 55.2 |
> | MAML | 79.6 | 85.1 | 55.5 |
> | FO-MAML | 79.1 | 84.8 | 55.4 |
> i-MAML | 79.2 | 85.0 | 55.9 |
> | Reptile | 79.2 | 84.2 | 55.4 |
> | ProtoNet | 79.8 | 84.3 | 55.2 |
> | ProtoNet-LITE | 79.2 | 84.4 | 55.3 |
> | LiFT K=1 (Ours) | 80.6 | 85.6 | 58.3 |
> | LiFT K=3 (Ours) | 80.6 | 85.7 | 59.1 |
> | LiFT K=5 (Ours) | 80.6 | 86.3 | 59.9 |
>
>
> > 2. The results do not include standard deviations. Are the improvements with LiFT statistically significant?
>
> Due to the computational overhead, it was time and compute-expensive to run multiple random seed experiments, and we only managed to run a single experiment. However, we are able to perform multiple 5 random seed runs on the Crossfit-CLS45 split to collect standard deviations and $p$-values. The results are shown in the following table ($p$-values of the existing methods against our LiFT-K=5 shown in the parentheses):
>
> |  | No Meta Train | Union Train | MAML | FO-MAML | i-MAML | Reptile |
> |:----:|:----:|:----:|:----:|:----:|:----:|:----:|
> | Score | $56.27\pm1.30$ | $56.79\pm0.59$ | $52.82\pm1.05$ | $57.70\pm1.18$ | $59.03\pm0.65$ | $58.26\pm0.66$ |
> | $p$-value | ($3.26 \times 10^{-6}$) | ($3.63 \times 10^{-8}$) | ($3.72 \times 10^{-8}$) | ($8.02 \times 10^{-6}$) | ($8.81 \times 10^{-7}$) | ($3.36 \times 10^{-7}$) |
>
> |  | BMAML M=5 | ABML | MAML-Mix K=5 | i-MAML-Mix K=5 | Retriever Mixture K=5 | Retriever Mixture K=all |
> |:----:|:----:|:----:|:----:|:----:|:----:|:----:|
> | Score | $57.98\pm0.59$ | $55.74\pm1.25$ | $53.87\pm0.12$ | $58.99\pm1.20$ | $55.82\pm1.42$ | $52.58\pm1.11$ |
> | $p$-value | ($1.09 \times 10^{-7}$) | ($1.47 \times 10^{-6}$) | ($7.19 \times 10^{-12}$) | ($4.85 \times 10^{-5}$) | ($4.06 \times 10^{-6}$) | ($4.87 \times 10^{-8}$) |
>
> |  | Retriever Fusion K=5 | Retriever Fusion K=all | MBC-$\mu$ K=5 | MBC-$\mu$ K=10 | LiFT K=5 (Ours) |  |
> |:----:|:----:|:----:|:----:|:----:|:----:|:----:|
> | Score | $56.03\pm1.69$ | $53.77\pm1.29$ | $56.17\pm1.05$ | $55.41\pm0.87$ | $63.87\pm0.31$ |  |
> | $p$-value | ($1.69 \times 10^{-5}$) | ($3.52 \times 10^{-7}$) | ($6.23 \times 10^{-7}$) | ($8.08 \times 10^{-8}$) | - |  |
>
> As shown, our LiFT model outperforms the existing methods statistically significantly.
>
> We have added this result in the revised paper (Appendix E.8 and Table 14).
>
>
> > 3. Some limitations, such as the joint training on the training tasks, which can be infeasible for large-scale datasets.
>
> We did mention this limitation in **Sec. 4.5 Utilizing Task-wise Pre-fine-tuned Models**. In this section, we also introduced a heuristic remedy of clustering pre-fine-tuned models. Table 5 shows the evaluation of this remedy, and the results are promising.
>
>
> > 4. Which architectures and hyperparameters are used?
>
> We used the same architectures and fair hyperparameters for MAML baselines (Appendix C). We agree that MAML is not scalable to large feature extractor networks. For fair comparison and for efficiency, we apply MAML in the same way that we apply LIFT. IE: We train MAML-of-LORAs, rather than MAML of the raw VIT-B/16 backbone, which would be intractable and unstable.
>
>
> > 5. Multiple inner steps for MAML.
>
> We have run the experiments to see the effect of the multiple inner loop updates in MAML. On the Cross-fit CLS-45 benchmark, increasing the number of inner iterations can improve the test performance at the cost of memory, but MAML still underperforms our LiFT by large margin.
>
> | # inner iters in MAML | 1 |  2 |  3 |  5 | LiFT (K=5) |
> |:----:|:----:|:----:|:----:|:----:|:----:|
> | CLS-45 | 52.68 | 48.48 | 55.73 | 56.02 | 64.12 |
>
> More than 5 iters in MAML incurred OOM issues.
>
> We have added this result in the revised paper (Appendix E.7 and Table 13).

---

> > ### Comment · Reviewer_pSzU · 2024-11-22
> > **Response to the authors' comments**
> >
> > I would like to thank the authors for addressing my concerns in the revised manuscript. I believe this is a good paper and will maintain my original score.

---

### Official Review · Reviewer_3TVT · 2024-11-03

**Soundness:** 3
**Presentation:** 3
**Contribution:** 3
**Rating:** 8
**Confidence:** 5

**Summary:**

The author proposed a modulated meta-learning scheme where the modulation is the LoRA parameter of the large model. Specifically, each task-specific parameter is represented with LoRA, and the base model is shared across all tasks. To learn such a model, the author proposed a hierarchical Bayesian meta-learning method called LiFT. Here, the task specifical LoRA is modeled to be sampled from a prior distribution, where the author have suggested an efficient sampling method.

**Strengths:**

The overall writing is clear, and the method itself is sensible.

The proposed sampling method is efficient. I think it would be great to show the experiment that shows the efficient gain.

**Weaknesses:**

Missing critical related works and comparison. Currently, there are many works that consider meta-learning with modulation (i.e., a few parameter updates from the base model, such as LoRA). For instance, CAVIA [1] is the first paper that suggested meta-learning with modulation. Furthermore, a more recent method, CNAPs [2], combines amortization-based meta-learning with modulation. Several works consider modulated meta-learning as follows: FiLM modulation with MAML [3,4], LoRA modulation with MAML [5], Scaling CNAPs to large-scale meta-learning [6,7], and LoRA modulation with amortization-based meta-learning [8].

[1] Fast Context Adaptation via Meta-Learning, ICML 2019\
[2] Fast and flexible multi-task classification using conditional neural adaptive processes, NeurIPS 2019\
[3] From data to functa: Your data point is a function and you can treat it like one, ICML 2022 \
[4] COIN++: Neural Compression Across Modalities, TMLR 2022\
[5] Modality-Agnostic Variational Compression of Implicit Neural Representations, ICML 2023\
[6] Memory Efficient Meta-Learning with Large Images, NeurIPS 2021\
[7] Improved Few-Shot Visual Classification, CVPR 2020\
[8] Online Adaptation of Language Models with a Memory of Amortized Contexts, arXiv 2024

----

Need to consider more recent baselines, and more effective meta-learning baselines. Currently, most of the meta-learning baselines are highly outdated. Furthermore, there are more effective and recent baselines [1,2,3]. Typically, [3] suggested the interpolation of sparse experts (i.e., only a few parameter updates), which has similarities with the current approach (i.e., LoRA modulation).

[1] Meta-learning with warped gradient descent, ICLR 2020\
[2] Bootstrapped meta-learning, ICLR 2022\
[3] Unleashing the Power of Meta-tuning for Few-shot Generalization Through Sparse Interpolated Experts, ICML 2024

---

I think the experiment application needs to be more motivating. The main purpose of using LoRA is to fine-tune large models to reduce the computation burden or overfitting. However, the current setup is mostly conducted in small-scale networks. I believe showing whether the proposed method scale to large-scale LLM (e.g., more than 1B param) will be an interesting and motivating example.

---

I agree that using meta-learning could be beneficial, but I don't understand the advantages of modeling with Bayesian meta-learning specifically. From an uncertainty perspective, it makes sense, but it's still possible to "jointly learn the source task" without a Bayesian approach. I'm particularly concerned about whether this proposed sophisticated sampling technique will truly scale with large models.

**Questions:**

See the question above.

---

> ### Author Response · Authors · 2024-11-21
> **Thank you for your valuable feedback!**
>
> > 1. Missing critical related works (8 papers [1-8]).
>
> In response to the reviewer’s suggestion, we have made an empirical comparison with the recent method LITE [6]. The rest of the models suggested are all tightly coupled with specific backbone network architectures, eg, CNAPs [2], and/or assume closed set classification problems e.g. [7]. As such they are not straightforwardly applicable to our main task of LoRA PEFT framework for adapting LLMs in open-ended text generation. But as we see that they are important latest meta learning algorithms, we have cited them in our revised paper (Please see Extended related work section in Appendix D, the second paragraph).
>
> The LITE [6] (memory efficient meta learning) is generally not tied to a specific network architecture, and as suggested therein, the ProtoNet meta learner can be applied. To this end, we ran ProtoNet-LITE on the VTAB benchmark since ProtoNet typically assumes small-way classification problems, not adequate for LLM’s large vocabulary output cardinality. LITE [6] essentially aims to represent the prototype vectors using the entire support data (instead of a minibatch), but to remedy computational overhead, they set a randomly chosen large portion of the support data as non-backpropagatable.
>
> The results are shown in the following table (the full scores for individual tasks are shown in Table 2 of the revised paper).
>
> |  | Non-natural -> Natural | Non-special -> Special | Non-structured -> Structured |
> |:----:|:----:|:----:|:----:|
> | No Meta Train | 78.0 | 84.3 | 54.4 |
> | Union Train | 79.0 | 84.5 | 55.2 |
> | MAML | 79.6 | 85.1 | 55.5 |
> | FO-MAML | 79.1 | 84.8 | 55.4 |
> i-MAML | 79.2 | 85.0 | 55.9 |
> | Reptile | 79.2 | 84.2 | 55.4 |
> | ProtoNet | 79.8 | 84.3 | 55.2 |
> | ProtoNet-LITE | 79.2 | 84.4 | 55.3 |
> | LiFT K=1 (Ours) | 80.6 | 85.6 | 58.3 |
> | LiFT K=3 (Ours) | 80.6 | 85.7 | 59.1 |
> | LiFT K=5 (Ours) | 80.6 | 86.3 | 59.9 |
>
>
> > 2. Need to consider more recent baselines, and more effective meta-learning baselines (3 papers [1,2,3]).
>
> [3] aims to learn a task-specific set of sparse masks for the new parameters added to the pre-trained ones while the new parameters are shared across the tasks (task-agnostic). Since they used ViT full backbone, and we are using the LoRA PEFTs, it is difficult to compare the two approaches directly. Only ViT-small was tested in [3] (sparse interpolation experts), and we also have ViT-B experiments.
>
> Instead we have implemented and run [2] bootstrapped-maml in our crossfit experimental framework with LoRA PEFT as parameters to learn. In the bootstrapped maml, the target model is first obtained by applying SGD updates on the train data then batch data sequentially. The meta learning loss is then the KL divergence between the predictive distributions of the train-data-updated model and the target model where the latter is treated as constant (no backprop).
>
> Comparison with “bootstrapped-maml” [2] on Crossfit CLS-45:
>
> |  | MAML | Bootstrapped MAML | LiFT (K=1) | LiFT (K=3) | LiFT (K=5) |
> |:----:|:----:|:----:|:----:|:----:|:----:|
> | CLS-45 | 52.68 | 55.02 | 62.96 | 63.37 | 64.12 |
>
> We have added this result in the revised paper (Appendix E.6, Table 12).
>
>
> > 3.  Scalability to large-scale LLM (e.g., more than 1B param).
>
> Our current results are already applied on VIT-B, as mentioned above, which is already larger scale than many meta-learning papers’ experiments. Nevertheless, we agree that it is interesting to explore scaling further.
>
> We now have run our algorithm on an even larger LLM, FLAN-T5-XL (3B), and the results on the Crossfit CLS-45 task split are summarized as follows. We ran all methods on a single A100 80GB GPU where iMAML (and its mixtures) and BMAML baselines incurred OOM.
>
> | CLS-45 | No-Meta-Train | Union-Train | MAML | FO-MAML | Reptile | LiFT K=1 (Ours) | LiFT K=3 (Ours) | LiFT K=5 (Ours) |
> |:----:|:----:|:----:|:----:|:----:|:----:|:----:|:----:|:----:|
> | FLAN-T5-XL (3B) | 65.13 | 73.41 | 73.74 | 74.22 | 69.39 | 78.37 | 79.69 | 80.01 |
> | BART-base | 57.34 | 57.25 | 52.68 | 59.38 | 57.59 | 62.96 | 63.37 | 64.12 |
>
> The result shows that our LiFT algorithm is scalable to an LLM with 3B parameters, where it continues to show large improvements over the baselines. Thanks for suggesting this experiment. We have added this result in our revised paper (Appendix E.1 and Table 6).
>
>
> > 4.  How about "jointly learning the source task" without a Bayesian approach?
>
> “Jointly learning the source task without a Bayesian approach” – This is exactly what “union-traininig” does, and it underperforms ours by large margin.

---

> ### Comment · Reviewer_3TVT · 2024-11-22
> **Thank you for the rebuttal**
>
> Thank you for the detailed response and clarification. My concerns are well-addressed and I have changed the score accordingly.

---

### Official Review · Reviewer_p6L8 · 2024-11-03

**Soundness:** 3
**Presentation:** 3
**Contribution:** 2
**Rating:** 6
**Confidence:** 3

**Summary:**

The authors propose a parameter efficient finetuning scheme called learning-to-finetune (LiFT) that can adapt a model not only to a single, but to a set of related tasks.

At the heart of LiFT sits a Bayesian meta training method that is executed on a set of related finetuing tasks. It uses hierarchical priors for the PEFT parameters to split task specific from task agnostic knowledge and runs stochastic gradient Langevin dynamics (SGLD) for posterior inference. The transferable task agnostic knowledge is later used as a prior for test-time adaptation.

While they explain their method, using LoRA, the method is general and can be adapted easily to any other PEFT scheme.

**Strengths:**

The paper is in general well written and I enjoyed reading it. Ideas are explained in detail. For most of the math intuitive explanations are provided. Hence, it is easy to understand the proposed meta-learning method and all the tricks that are needed to make it work.

The paper contains two creative ideas: 1) Their particular hierarchical Bayesian model for the PEFT parameters. 2) The combination of Gibbs sampling and SGLD for memory efficient posterior inference.

**Weaknesses:**

There are several weaknesses to this paper:
* The main claim of the paper is, that it is beneficial to use their Bayesian formulation for PEFT of models to a set of related tasks. They give an intuitive explanation that one of their latent variables learns task agnostic and the other task specific adaptations. While this sounds reasonable, there is no analysis if this claim is actually true and, hence, if their hierarchical model actually makes sense. However, experimental sanity checks could be easily made e.g. by comparing to a non-hierarchical model. In the end I am completely puzzled what part causes the demonstrated increase in performance: 1) The formulation of the problem or 2) favorable training dynamics of this quite large training algorithm.

* Important ablation studies are not provided: Their hierarchical model requires the choice of two variances $\sigma^2$ and $\beta^2$. Only one set of values is provided. However, to judge how brittle the model is, some ablations would be helpful.

* The paper does not give any idea how the proposed algorithm scales with #adapted parameters (can it only be used with PEFT methods or even with FFT?). Training requires to run the SGLD sampling. From the appendix I got, that it requires quite some steps, i.e., 2000 burn-in and 1000 warmup steps. It is not obvious how this translates to training time.

* The paper proposes an online EM algorithm to fit a GMM to posterior samples in an iterative way, making it unnecessary to store the full set of posterior samples. While stating that this is a novel contribution, there exists lots of work about this problem already. Keywords are: 1) incremental EM or 2) streaming GMMs [1], [2]. Related works are not referenced and a comparison is missing.

[1] Hosseini, Reshad, and Suvrit Sra. "An alternative to EM for Gaussian mixture models: batch and stochastic Riemannian optimization." Mathematical programming 181.1 (2020): 187-223.
[2] Karimi, Belhal, et al. "On the global convergence of (fast) incremental expectation maximization methods." Advances in Neural Information Processing Systems 32 (2019).

**Questions:**

After thoroughly reading the paper, some questions remain:
* Is there any experimental evidence that $\phi$ really learns meaningful task-agnostic adaptations?

* How brittle is the training if we change the parameters $\sigma^2$ and $\beta^2$ of the hierarchical model? Did you do any experiments there?

* Why do you choose a GMM to model $\phi|\{D_i\}_{i=1}^N$.

I think, having a multi-modal distribution goes against your idea that $\phi$ learns task-agnostic adaptations, because each mode can represent a specialization. More specifically, if the number of modes $M$ is equal to the number of tasks $N$, each mode of $\phi|\{D_i\}_{i=1}^N$ can specialize to one task. In this case you would have something very similar to a stochastic version of the mixtures of LoRA idea. How do you prevent this from happening?

* After test-time adaptation, is the model output stochastic or deterministic? I.e. do you continue to run the SGLD sampling for $\theta|\phi$ or do you use some statistics and perform just a deterministic forward pass?

* In the stochastic case: Why don't you provide confidence intervals for the LiFT results?

* How does the LiFT scale with the #trainable parameters?

* What is the difference between warm-up samples and burn-in samples for the SGLD Gibbs sampling?

I am looking forward to your explanations.

---

> ### Author Response · Authors · 2024-11-21
> **Thank you for your valuable feedback! (Part 1)**
>
> > 1. Why the hierarchical Bayesian method works.
>
> To show the importance of hierarchical modeling, we have implemented and tested two *non-hierarchical* Bayesian models. In the first model, the PEFT LoRA parameters are treated as random variables as ours, but they are shared across all tasks. Hence the posterior inference in this case amounts to learning  task-shared (task-agnostic) information from meta training data. We used the SGLD posterior inference. This model performs way worse than our hierarchical model as shown in the table below.
>
> As a second non-hierarchical Bayesian model, we can think of a model where each task is represented by its own PEFT LoRA parameters, but there is no governing higher-level random variables as in our LiFT model. Consequently this model aims to learn only task-specific information, and would not transfer anything to novel tasks. Hence it can be seen as a Bayesian version of "No Meta Train" method, which only runs test-time adaptation via SGLD posterior inference. This model also performs poorly compared to our hierarchical model as shown in the following table.
>
> |  | Non-hierarchical (Shared PEFT) | Non-hierarchical (Task-wise PEFT) | LiFT (K=1) | LiFT (K=3) | LiFT (K=5) |
> |:----:|:----:|:----:|:----:|:----:|:----:|
> | CLS-45 | 55.62 | 56.74 | 62.96 | 63.37 | 64.12 |
>
> We have added this result in the revised paper (Appendix E.2 and Table 7).
>
>
> > 2. Need ablation study on $\sigma^2$ and $\beta^2$.
>
> To see the robustness to these hyperparameters, we have run the ablation study with our LiFT $K=5$ model on the Crossfit CLS-45 task split. As shown in the following table, the test performance is quite robust to the choice of ($\sigma^2$, $\beta^2$).
>
> ($\sigma^2,\beta^2$) | 0.001 | 0.005 | 0.01 (reported) | 0.05 | 0.1 | 0.2 | 0.3 | 0.5 |
> |:----:|:----:|:----:|:----:|:----:|:----:|:----:|:----:|:----:|
> | Score | 62.87 | 63.05 | 64.12 | 63.62 | 62.75 | 62.75 | 61.48 | 61.73 |
>
> Please see also the plot of this ablation study in our revised paper (Appendix E.3 and Figure 7).
>
>
> > 3. How the proposed algorithm scales with #adapted parameters (can it only be used with PEFT methods or even with FFT?). Training times for the warm-up and burn-in steps.
>
> There is no point in adapting the full backbone parameters (FFT), which is known to overfit and underperform PEFTs in situations with modest downstream task sizes.
>
> We report the wall clock training times (on a single V100 GPU) in the following table. Our LiFT with $K=5$ GMM is compared with the two baselines “Union Train” and “MAML” on two different PEFT (rank 4 and 64) cases. The result below signifies that our model is as efficient as the vanilla SGD update (ie, “Union Train”) for both low and high ranks, thanks to our efficient SGLD-Gibbs sampler.
> Asymptotically (in Big-$O$ notation), both the vanilla SGD update and our SGLD-Gibbs (Eq.9-11) $+$ online EM (Eq.13-14) take the same $O(T+d)$ time per iteration where $T$ is the time for the backbone forward/backward computation for evaluating $\nabla_{\theta_i^a} \log p(D_i|\theta_i^a)$ and $d$ is the number of PEFT parameters. This is because of the Gaussian $p(\theta_i^a|\phi)$ which allows analytic gradient $\nabla \log p(\theta_i^a|\phi)$ with respect to both $\theta_i^a$ and $\phi$. The EM iteration also takes $O(d)$ time.
>
> |  | Rank 4 LoRA | Rank 64 LoRA |
> |:----:|:----:|:----:|
> | (LiFT K=5) Per-task/iter time (after burn-in) | 0.268 secs | 0.287 secs |
> | (“Union Train”) Per-task/iter time | 0.242 secs | 0.270 secs |
> | (“MAML”) Per-task/iter time | 0.372 secs | 0.380 secs |
>
> The warm-up steps in our method refers to the vanilla SGD steps for the first 1000 iterations, and the burn-in steps amounts to running the proposed SGLD recurrences for the next 1000 iterations (without collecting the posterior samples). After the burn-in steps, we collect the posterior samples while running the SGLD. The wall clock times for these two stages are shown below, and they are quite reasonable times.
>
> |  | Rank 4 LoRA | Rank 64 LoRA |
> |:----:|:----:|:----:|
> | (LiFT K=5) Warm-up time (1~1000th steps) | 219 secs | 233 secs |
> | (LiFT K=5) Burn-in time (1001~2000-th steps) | 277 secs | 279 secs |
>
> We have added the above results in the revised paper (Appendix E.5, Table 10 and 11).
>
> *(Our responses continue in the thread below.)*

---

> > ### Author Response · Authors · 2024-11-21
> > **Part 2**
> >
> > > 4. Existing works on online EM algorithm.
> >
> > We realize that there have been many prior works on online EM methods in the literature. We have cited them along with the papers suggested by the reviewer in the extended related work section Appendix D in the revised paper. Although we find that none of them are identical to the one proposed in our paper, most are similar in nature to ours (e.g., exploiting the recursive structure of the EM). Hence we believe that these prior online EM methods can be employed in our online posterior estimation to be equally successful.
> >
> > The first paper (Hosseini and Sra 2020) suggested by the reviewer, is actually not very relevant to our work, since they rather proposed an SGD learning of the GMM on a Riemannian manifold, showing that it outperformed the conventional EM algorithm. Hence not quite related to the online EM.
> >
> > In the second paper (Karimi, Belhal, et al. 2019), they analyzed incremental and stochastic versions of the EM algorithm as well as extending the variance reduction technique in a common unifying framework.
> >
> >
> > > 5.  Why do you choose a GMM to model $p(\phi | \{D_i\}_{i=1}^N)$?
> >
> > Because the posterior distribution is expected to be multi-modal. Various recent papers on the mixtures of PEFTs (as cited in our paper) rely on the same idea of the effectiveness of the multiple underlying PEFT prototypes.
> >
> >
> > > 6. If the number of modes is equal to the number of tasks $N$, each mode of $\phi | \{D_i\}_{i=1}^N$ can specialize to one task. In this case you would have something very similar to a stochastic version of the mixtures of LoRA idea. How do you prevent this from happening?
> >
> > We pre-specify the number of clusters, much smaller than $N$, as an inductive bias. If the number of clusters becomes equal to $N$, we agree that the model would overfit and not extract any task agnostic information. But this would not be a reasonable hyperparameter to set as firstly it would be slower to run; and secondly – as per any mixture model – the number of mixture components should be much less than the number of data (=tasks in our case).
> >
> > Overall there is a tradeoff. With a uni-modal posterior for $\phi$ the hierarchical model exacts fully task agnostic information, but it may underfit by forcing dissimilar tasks to be too similar (e.g., LIFT K=1 or  ABML). With a multi-modal posterior and $K=N$ there may be no extraction and transfer of task-agnostic knowledge. By learning a multi-modal posterior with $1<K<N$ we group similar tasks into common clusters and share knowledge between them, while requiring no knowledge transfer between dissimilar tasks in different clusters that would lead to negative transfer/underfitting.
> >
> > This high-level notion of clustering similar tasks and enforcing similarity between tasks in the same cluster was also used in other related work that we compare against such as Lora-Mixture and MAML-mixtures (Tab 1).
> >
> >
> > > 7. After test-time adaptation, is the model output stochastic or deterministic?
> >
> > We use the last sample $\theta_*^a$ from the SGLD run Eq.(18) in the inference.
> >
> >
> > > 8. Why don't you provide confidence intervals for the LiFT results?
> >
> > We have also collected 4 more SGLD samples $\theta_*^a$ from Eq.(18) in addition to the last one.  Using these 5 samples we have taken the MC average during the test time predictive distribution computation. This gives us some confidence intervals. For the CLS-45, K=5 LiFT, we have **$63.81 \pm 0.40$** (64.12 reported in the paper using the last SGLD sample).
> >
> >
> > > 9. What is the difference between warm-up samples and burn-in samples for the SGLD Gibbs sampling?
> >
> > Warm-up stage means before starting to run SGLD steps, just a regular $\theta$ training (without $\phi$ update). After the warm-up stage, we have the burn-in stage in which we start running the SGLD steps Eq.(9,10,11), but do not collect the samples (and no posterior GMM maintenance). After the burn-in stage, we start collecting posterior samples and build/update the GMM.

---

> > > ### Author Response · Authors · 2024-11-26
> > > **To Reviewer **p6L8****
> > >
> > > We hope that you have had a chance to read our rebuttal. We would really appreciate it if you let us know if we answered your questions and concerns properly before the discussion period ends.

---

> > > > ### Comment · Reviewer_p6L8 · 2024-11-26
> > > > **Thank you**
> > > >
> > > > Thank you for the clarifications and additional experiments. All my doubts have been adressed. I raised my score.

---

### Official Review · Reviewer_PRGJ · 2024-11-05

**Soundness:** 3
**Presentation:** 3
**Contribution:** 3
**Rating:** 8
**Confidence:** 4

**Summary:**

The paper develops a general fine-tuning strategy for foundational models that is based on the meta-learning principle. This way not only the foundational models become shareable, but also the layers responsible for task-specific fine-tuning can be accumulated and shared to be used on unseen tasks with not enough fine-tuning data. The core contribution of the paper is the Bayesian methodology that casts meta-learning (and hence task-specific fine-tuning) as Bayesian sampling exercise. Technically, this is implemented via a modification of SGLD and the online EM algorithm. Empirical results are obtained from NLP and Vision tasks.

**Strengths:**

- Overall, the idea seems sufficiently novel and interesting
- The empirical evaluation is extensive and convincing
- Results are state of the art

**Weaknesses:**

- The justification of SGLD-Gibbs Sampling is only empirical through a toy example in Appendix A. A theoretical justification showing the required convergence would have significantly strengthened the contribution.
- Ablation studies are not comprehensive enough, failing to support major decisions made in the algorithm design step: the $J$-term update and the Online-EM Mixture for Posterior Approximation. See questions section for details.
- It looks like literature review could be updated with the online EM literature. I am not sure if the online EM algorithm is genuinely novel given the lack of analysis of the related work on this topic in the paper. See questions for details.
- It is unclear if code is being open sourced.

**Questions:**

- $J$ appears in (11) out of nowhere. Could you please motivate the need in this term more clearly in the text transition between (7) and (9)? It seems to be the core algorithmic contribution that does not follow trivially from the original SGLD formulation in (7-8). It feels like by not discussing it in sufficient detail authors basically undersell their contribution.
- Can you say at least something theoretical about your approximations, to strengthen the theory? I understand that convergence rate analysis might be to much of an ask, but if we talk about means and if you take the expectations of (9-11) will they match the expectations of equations (7-8) in the limit?
- On a related note, can you run an ablation study by comparing against a more naive version of the algorithm that does not rely on $J$ updates? For example, you could use the sample approximation of the sum in (7) by the log-probability of $\theta_{i}^a$ in a given update round. In my view, this could further strengthen the algorithmic contribution of the paper, or simplify the algorithm if sample approximation has same or better accuracy.
- "However, we aim to enrich it by a mixture of Gaussians to better approximate the true posterior that is inherently a multi-modal distribution". Can you provide the results of ablation study comparing against Gaussian confirming that such enrichment is actually useful? This is especially important given that the Online-EM Mixture for Posterior Approximation is necessary only to support the GMM. If the GMM is not provably necessary, the value of the Online-EM Mixture contribution is questionable.
- Since you are claiming online EM as a technical contribution, could you please update the related work section with the online EM literature? For example: https://arxiv.org/abs/2207.14019, https://www.diva-portal.org/smash/get/diva2:857377/FULLTEXT01.pdf, https://www.sciencedirect.com/science/article/abs/pii/S0167947304003263. Please discuss the novelty of your work w.r.t. existing contributions and motivate why you need a new version of online EM.
- "Hence we stick to the simple average of the metrics over all test tasks regardless of the metric types." I understand that the relative lift could be tricky, because of division by small numbers. Could you please consider reporting the mean of the absolute deltas between the baseline and the candidate algorithm? I believe this could be a more robust and more statistically significant measure of accuracy improvement than reporting the sum of raw metric values.
- Will code be open-sourced?

---

> ### Author Response · Authors · 2024-11-21
> **Thank you for your valuable feedback!**
>
> > 1. A theoretical justification of the SGLD-Gibbs
>
> The convergence of the SGLD recurrences to the target distribution has been studied recently in (Zou et al., 2021, Xu et al., 2018, Raginsky et al., 2017). Also it is well known in the MCMC literature that the Gibbs sampling, if all the variables are visited sufficiently many times, converges to the stationary distribution since it satisfies both the detailed balance equation and ergodicity. Then since our SGLD-Gibbs has the transition kernel in MCMC that is a composition of these two operators, it should converge to the target distribution.
>
> We have added the above argument in our revised paper (Footnote 1 in p.4).
>
> However, what is not evident in theory and needs further theoretical investigation, is whether our asynchronous update scheme in Eq.(9-11) would also converge to the stationary distribution of the chain. This is mainly because the sum of the gradients $\nabla_\phi \log p(\theta_i^a|\phi)$ over all tasks $i$ in the $\phi$ update Eq.(7) is not exact in the asynchronous scheme since we use the cavity sum (ie, the sum except the current task $i$) that is computed from the old $\phi$s.
>
> We will be investigating further this theoretical analysis, but it may be difficult for us to come up with a decent theoretical conclusion during this rebuttal stage.
>
>
> > 2. An alternative version to the $J$ update scheme. Eg, sample approximation of the sum in (7).
>
> To see the impact of our $J$ update strategy (Eq. 9-11), we have run the stochastic approximate version of (7), which replaces the average of $\nabla_\phi \log p(\theta_i|\phi)$ by the current task iterate alone. As shown in the table below, it has decent performance but slightly underperforms our original proposal of the $J$ update strategy. We thank the reviewer for an interesting suggestion, and we have mentioned it in the revised paper (Please see Appendix E.4 and Table 9).
>
> |  | "Sample approximation" | "$J$ update" (our original approach) |
> |:----:|:----:|:----:|
> CLS-45 | 63.61 | 64.12 |
>
>
> > 3. Related work section with the online EM literature.
>
> We realize that there have been many prior works on the online EM methods in the literature. We have cited them along with the papers suggested by the reviewer in the extended related work section Appendix D in the revised paper. Although we find that none of them are identical to the one proposed in our paper, most are similar in nature to ours (e.g., exploiting the recursive structure of the EM). Hence we believe that these prior online EM methods can be employed in our online posterior estimation, and can be equally successful.
>
> The discussions on the related online EM algorithms can be found in Appendix D in the revised paper.
>
> > 4. Code open sourcing.
>
> We consider releasing our code should the paper be accepted.
>
>
> > 5. Motivation on the $J$ update scheme (Need more elaboration).
>
> Although $J$ is actually introduced in L:179 p.4 where we stated that $J$ represents and maintains $\sum_i \nabla_\phi \log p(\theta_i|\phi)$, we now elaborate this further in the revised paper. Please see the text before Eq.(9). We reiterate it as follows:
>
> Another computational bottleneck is the sum of the gradients $\nabla_\phi \sum_i \log p(\theta_i^a|\phi)$ over all $i=1,\dots,N$ for each $\phi$ update in (7). To remedy this issue, we introduce an auxiliary variable $J$ that maintains this sum of the gradients, and we let it asynchronously updated (11): at each iteration for task $i$, subtracting the old gradient for $i$ from $J$ to approximate the cavity sum $\sum_{i'\neq i} \nabla_\phi \log p(\theta_{i'}^a|\phi)$ and adding a new gradient $\nabla_\phi \log p(\theta_{i}^a|\phi)$ to $J$.
>
>
> > 6. Provide results of ablation study comparing against Gaussian confirming that such enrichment is actually useful.
>
> Fig. 3 already shows comparison between LiFT Gaussian posterior approximation (K=1) vs. LiFT GMM posterior approximation (K=3 and K=5). Clearly, GMM is better than Gaussian (K=1) for most cases.
>
>
> > 7. Report means of the absolute deltas between the baseline and the candidate algorithm.
>
> We report the mean of absolute deltas on the CLS-45 task split as follows:
>
> |  | No-Meta-Train | Union-Train | MAML | FO-MAML | i-MAML | Reptile
> |:----:|:----:|:----:|:----:|:----:|:----:|:----:|
> MAD score | 8.79 | 11.27 | 13.10 | 6.80 | 7.04 | 9.11 |

---

> > ### Comment · Reviewer_PRGJ · 2024-11-28
> > **Post rebuttal response**
> >
> > I thank the authors for their detailed response that addresses my concerns. I will raise my score accordingly.

---

### Meta-Review · Area_Chair_miKZ · 2024-12-18

**Metareview:**

This paper proposes a hierarchical Bayesian meta-learning framework for parameter-efficient fine-tuning (PEFT). The key idea is to model LoRA parameters as latent variables drawn from higher-level, task-agnostic variables, thereby enabling flexible knowledge sharing across tasks. To handle the complexity and scale of the posterior inference, the authors introduce a SGLD-Gibbs sampling algorithm and an online EM approach to approximate the posterior distribution efficiently.

All reviewers agreed that the paper offers a novel and principled approach to meta-learning with PEFT. Initial concerns included the absence of comparisons with more recent meta-learning methods, limited theoretical discussion regarding the asynchronous update scheme, and a lack of certain ablation studies. The authors addressed these points thoroughly during the rebuttal phase. After these clarifications and additional experiments, all reviewers ultimately recommended acceptance.

After reading the paper, reviews, and discussions, the AC believes that the paper makes a strong and timely contribution. By extending the idea of PEFT to a hierarchical Bayesian meta-learning framework, it provides a principled way to decompose LoRA parameters into task-agnostic and task-specific components. The experiments confirm the scalability and effectiveness of the proposed method.

AC encourages the authors to incorporate the additional results and discussions from the rebuttal—especially those involving stronger meta-learning baselines and more comprehensive ablation studies—into the final version of the paper.

**Additional Comments On Reviewer Discussion:**

During the discussion, Reviewer PRGJ raised concerns about theoretical and empirical validation for the asynchronous SGLD-Gibbs updates. Reviewer p6L8 emphasized the need for further ablations, including checks on the necessity of the hierarchical Bayesian framework, the robustness of chosen hyperparameters, and the utility of a Gaussian mixture rather than a single Gaussian. Reviewer 3TVT requested additional comparisons with more recent meta-learning methods and confirmation of scalability to larger models. Reviewer pSzU sought statistical significance assessments and clarifications on baseline configurations.

In response, the authors conducted new experiments to compare LiFT against non-hierarchical baselines, demonstrated that their mixture-based posterior representation outperforms simpler alternatives, offered sensitivity analyses for key hyperparameters, and included comparisons with stronger baselines. These steps addressed the core concerns, convincing the reviewers to suggest acceptance.

---

### Decision · Program_Chairs · 2025-01-22

Accept (Spotlight)